# Neuronal morphologies built for reliable physiology in a rhythmic motor circuit

Adriane G Otopalik[1,2†]*, Jason Pipkin[1], Eve Marder[1]*

[1]Volen Center and Biology Department, Brandeis University, Waltham, United States; [2]Grass Laboratory, Marine Biological Laboratories, Woods Hole, United States

**Abstract** It is often assumed that highly-branched neuronal structures perform compartmentalized computations. However, previously we showed that the Gastric Mill (GM) neuron in the crustacean stomatogastric ganglion (STG) operates like a single electrotonic compartment, despite having thousands of branch points and total cable length >10 mm (Otopalik et al., 2017a; 2017b). Here we show that compact electrotonic architecture is generalizable to other STG neuron types, and that these neurons present direction-insensitive, linear voltage integration, suggesting they pool synaptic inputs across their neuronal structures. We also show, using simulations of 720 cable models spanning a broad range of geometries and passive properties, that compact electrotonus, linear integration, and directional insensitivity in STG neurons arise from their neurite geometries (diameters tapering from 10-20 μm to $\leq 2$ μm at their terminal tips). A broad parameter search reveals multiple morphological and biophysical solutions for achieving different degrees of passive electrotonic decrement and computational strategies in the absence of active properties.

DOI: https://doi.org/10.7554/eLife.41728.001

**\*For correspondence:**
aotopali@brandeis.edu (AGO);
marder@brandeis.edu (EM)

**Present address:** †Department of Biological Sciences, Columbia University, New York, United States

## Introduction

Neurons often present complex and highly-branched morphologies. How synaptic voltage events propagate within and across neurite branches is determined by the structure's geometrical and biophysical properties. Passive voltage propagation is influenced by the neurite's diameter, membrane resistance, and axial resistance (**Rall, 1959**; **Rall, 1960**; **Rall, 1969**; **Jack et al., 1975**; **Holmes, 1989**). Rall and colleagues were the first to apply passive cable theory and develop the equivalent cylinder model for the study of electrotonus in single dendrites and branched dendritic trees (**Rall, 1959**; **Rall, 1960**; **Rall, 1969**). Schierwagen then devised a broader mathematical description of membrane voltage distributions in complex, highly-branched neurite trees with non-uniform boundary conditions and geometries (**Schierwagen, 1989**). These seminal theoretical studies have provided an invaluable framework for understanding passive neuronal physiology. However, in a functioning neuron, synaptic inputs, receptors, and ion channels may shunt or amplify propagating voltage signals (**London and Häusser, 2005**). Thus, the transformation from neuronal morphology to electrophysiological activity patterns is often unpredictable in the absence of direct experimental assessment. To date, measuring voltage attenuation across the many neurite paths presented in complex neuronal structures using electrophysiological techniques has proven difficult. Thus, electrotonus has been experimentally assessed in only a handful of neuron types (for example: **Spruston and Johnston, 1992**; **Spruston et al., 1994**; **Rapp et al., 1994**; **Carnevale et al., 1997**; **Stuart and Spruston, 1998**; **Chitwood et al., 1999**; **Jaffe and Carnevale, 1999**; **Otopalik et al., 2017b**; **Medan et al., 2018**), and this greatly restricts our understanding of the breadth of biophysical organizations utilized in different neuron types and circuit contexts.

In two recent studies, we characterized the morphology (*Otopalik et al., 2017a*) and passive electrophysiology (*Otopalik et al., 2017b*) of the identified neurons of the crustacean stomatogastric ganglion (STG), a small central pattern-generating circuit mediating the rhythmic contractions of the animal's foregut. The 14 identified neurons of the STG present distinct, cell-type-specific electrophysiological waveforms, firing patterns and circuit functions (*Harris-Warrick et al., 1992*). We quantified numerous morphological features pertaining to the macroscopic branching patterns and fine cable properties of four neuron types (*Otopalik et al., 2017a*). Interestingly, the four neuron types did not adhere to optimal wiring principles (*Cuntz et al., 2010*) or Rall's 3/2 rule (*Rall, 1959*) and exhibited expansive neurite trees that sum to >10 mm of total cable length, tortuous and long individual branches (ranging between 100 μm and 1 mm in length) and thousands of branch points with complex geometries. There was quantifiable inter-animal variability in many features within neuron types, and no single metric or combination of metrics distinguished the four neuron types.

In a second study, we then asked: How do STG neurons produce reliable firing patterns across animals, given their apparently inefficient and highly variable structures? As a first examination of how neuronal morphology maps to physiology in the STG, we characterized electrotonus, or passive voltage signal propagation, in one STG neuron type, the Gastric Mill (GM) neuron. We were surprised to find that, despite their expansive and complex neuronal structures, GM neurons are relatively electrotonically compact and operate much like single electrical compartments (*Otopalik et al., 2017b*). We suggested that compact electrotonic structures may effectively counteract the potential physiological consequences of morphological variability observed in GM neurons across animals (*Otopalik et al., 2017a*; *Otopalik et al., 2017b*).

We first motivate the present study with an empirical numerical model that measures electrotonus in a library of cable models with varying geometrical and passive properties. In doing so, we recapitulate passive electrotonic decrement described in the aforementioned seminal theoretical studies (*Rall, 1959*; *Rall, 1960*; *Rall, 1969*). Yet, we also show that a subset of cable models with geometries consistent with those observed in multiple STG neuron types (*Otopalik et al., 2017a*), are relatively resilient to electrotonic decrement. Given this prediction, we asked whether compact electrotonus is a generalizable feature in the STG, and therefore common among multiple, distinct neuron types. Using glutamate photo-uncaging in tandem with intracellular electrophysiology (as in *Otopalik et al., 2017b*), we measure electrotonus in four STG neuron types. We then investigate how neurite geometry shapes voltage integration in these complex neuronal structures. By complementing these experiments with validating computational simulations, we demonstrate that STG neurite geometries and passive properties are sufficient to account for the compact electrotonus and voltage integration observed in this pattern-generating circuit. Furthermore, our numerical simulations suggest that different neurite geometries may be suitable for circuits that subserve different functions.

## Results

### Simulating electrotonus in diverse neurites

In previous work, we were surprised to find that STG neurons exhibit neurite lengths as long as 1 mm and neurite diameters between 10–20 μm at their primary neurite junctions, that decrement to ≤1 μm at their terminating tips (*Otopalik et al., 2017a*), suggesting geometries with a 1–2% taper. Passive cable theory suggests that voltage signals passively propagating such long distances are likely to undergo a great deal of attenuation (*Rall, 1960*; *Rall, 1964*; *Rall, 1969*; *Jack et al., 1975*; *Schierwagen, 1989*). Experimental studies to date have validated predictions presented in these theoretical studies in dendrites with uniform and/or smaller diameters (typically ≤3 μm; *Holmes, 1989*; *Jaffe and Carnevale, 1999*; *Stuart and Spruston, 1998*). These studies suggested that diameter is a critical parameter in determining electrotonic decrement in the absence of amplifying or shunting mechanisms. Yet, none of these studies examined electrotonus in neurites with the wide diameters exhibited by STG neurites. Thus, we first asked whether there may be a boundary at which a neurite's diameter is wide enough to overcome distance-dependent attenuation.

We conducted a proof-of-concept computational characterization of electrotonus in a library of 720 passive cable models with a much broader range of geometries (*Figure 1A*) and passive properties (specific $R_a$ values between 50–300 Ω x cm and specific $R_m$ values between 1,000–20,000 Ω x cm$^2$; see Materials and methods) than explored in earlier studies. These geometries range from narrow cables with uniform diameters (*Figure 1A*, gray) to broad cables with diameters reminiscent of STG neurites (*Figure 1A*, blue), thereby spanning the broad range of neurite geometries existing in diverse nervous systems. Electrotonus was characterized by measurement of an effective electrotonic length constant ($\lambda_{effective}$ in µm; see Materials and methods), which is equivalent to the distance at which a voltage signal decrements to 37% of the maximal voltage amplitude (at the activation site). Thus, greater $\lambda_{effective}$ values are suggestive of less electrotonic decrement. *Figure 1B* illustrates the measurement of $\lambda_{effective}$ in a classic cable model with a uniform diameter of 0.5 µm, $R_a$ = 100 Ω x cm, $R_m$ = 10,000 Ω x cm$^2$. In this case, the voltage event attenuates greatly with distance ($\lambda_{effective}$ is approximately 300 µm). This voltage attenuation is robust and $\lambda_{effective}$ is less than 1 mm for a broad range of passive properties (*Figure 1C*). Altering the morphology of this cable model to reflect the geometry of an STG neurite (*Figure 1D*) results in a smaller response amplitude at the site of activation (compare maximum amplitudes in *Figure 1B and D*). However, this is accompanied by a robust increase in $\lambda_{effective}$ and similar signal amplitudes across the cable. Larger $\lambda_{effective}$ values are observed for a broad range of passive parameters (*Figure 1E*). $R_a$ values between 50–150 Ω x cm and $R_m$ values $\geq$ 10,000 Ω x cm$^2$ yield $\lambda_{effective}$ values greater than 1 mm. Examining the entire morphological and biophysical parameter space (*Figure 1—figure supplement 1*) demonstrates that a range of $\lambda_{effective}$ values can be achieved across neurites varying not only in their passive properties, but also in their geometries. Neurites with wide proximal diameters present relatively long $\lambda_{effective}$ values even in the presence of a large load at the proximal end of the cable ($d_0$), made to mimic a putative shunt imposed by the rest of the neurite tree (*Figure 1—figure supplement 2*). Importantly, these simulations demonstrate that long $\lambda_{effective}$ values can be achieved as a consequence of neurite geometry alone, in the absence of voltage-gated ion channels or other amplifying mechanisms that have been implicated in boosting distally-evoked events in other neuron types (*Magee and Cook, 2000*; *Andrasfalvy and Magee, 2001*; *Smith et al., 2003*; *Gulledge et al., 2005*; *Lavzin et al., 2012*).

## Linking electrotonus and neurite geometry in STG neurons

Characterization of electrotonus in our cable model library suggests that the wide diameters of STG neurites may effectively equalize passive voltage signal propagation. Consistent with this prediction, we recently demonstrated that the Gastric Mill neuron exhibits a relatively uniform electrotonic structure (*Otopalik et al., 2017b*). To determine if this is a generalizable feature across different STG neuron types, we characterized electrotonus in multiple neurites of three additional STG neuron types: Lateral Pyloric (LP), Ventricular Dilator (VD), and Pyloric Dilator (PD) neurons, while also corroborating our previous findings in GM neurons (N = 5–6 neurons of each type). These four neuron types present equally complex and expansive morphologies, but distinct voltage waveforms and circuit functions (*Figure 2*). PD, LP, VD, and GM neurons were unambiguously identified by their innervation patterns (*Figure 2A*) and by matching their intracellular spiking patterns with concurrent extracellular recordings of nerves known to contain their axons (*Figure 2B*). PD and LP neurons innervate two muscles in the pylorus of the foregut (*Figure 2A*) and participate in the ongoing, triphasic pyloric rhythm (*Figure 2B*). PD and LP can both be identified by matching their intracellular firing patterns with spiking units on the lateral ventricular nerve (*lvn; Figure 2B*). The VD neuron innervates the cv1 muscle of the pylorus and can be identified on the medial ventricular nerve (*mvn; Figure 2B*). The GM neuron participates in the episodic gastric mill rhythm, innervates gm1, 2, and 3 muscles (*Figure 2A*), and can be identified on the dorsal gastric nerve (*dgn; Figure 2B*). When filled with fluorescent dye, each neuron presents highly branched and expansive neurite trees (*Figure 2C*). The morphological features of these neuron types have been described quantitatively and in detail in previous studies (*Wilensky et al., 2003*; *Bucher et al., 2007*; *Thuma et al., 2009*; *Otopalik et al., 2017a*). To examine neurite geometries in these four neuron types, we completed volumetric reconstructions and continuous measurement of neurite diameters (from soma to terminating tip) for 23 neurite paths, for which we had confocal image stacks with sufficient resolution. Recapitulating previous work, we observed neurite paths that taper from 10 to 20 µm at the soma-primary neurite

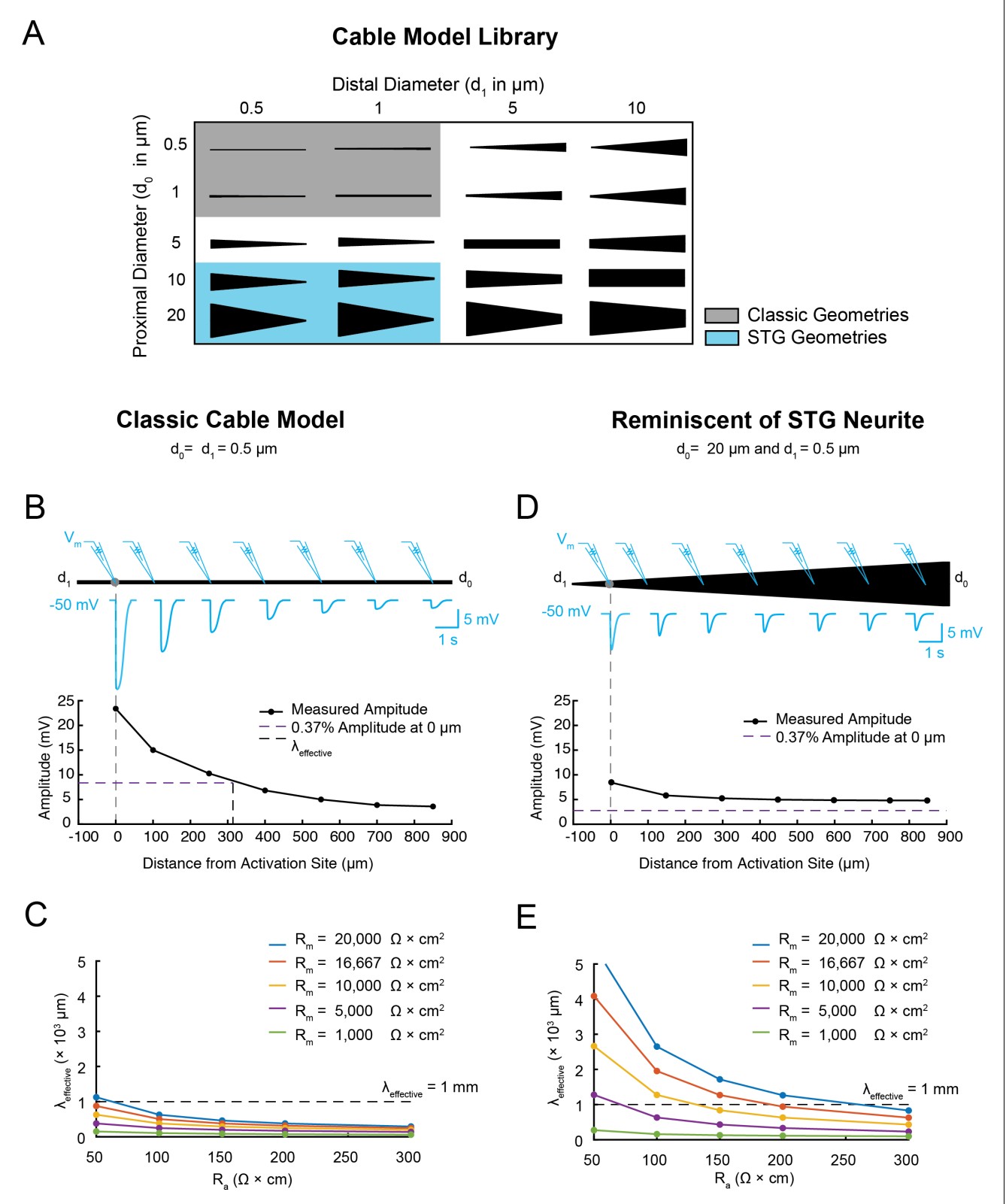

**Figure 1.** Electrotonus in cable models with diverse passive properties and geometries. (**A**) Illustration of the complete matrix of cable model geometries assessed in the computational simulation. Proximal diameters ($d_0$) ranged between 0.5–20 μm and distal diameters ($d_1$) ranged between 0.5–10 μm, yielding 20 different geometries spanning from fine, uniform-diameter cylinders to immensely tapered cables. Shaded areas indicate geometries consistent with vertebrate neocortical and hippocampal pyramidal neuron dendrites (gray) and neurites of STG neurons (blue). (**B**) Top:

*Figure 1 continued on next page*

*Figure 1 continued*

illustration depicting simulated measurement of the effective electrotonic length constant ($\lambda_{effective}$) in a classic cable model with a uniform 0.5 μm-diameter ($R_m$ = 10000 Ω*cm$^2$ and $R_a$ = 100 Ω*cm). An inhibitory potential ($E_{rev}$ = −75 mV, τ = 70 ms, $g_{max}$ = 10 nS) was evoked at a distal site (gray circle) and recorded (blue traces) at increasing distances from the site of activation (0, 100, 250, 400, 550, 700, 850 μm). Bottom: Plot depicts the amplitude of the evoked inhibitory potential measured at increasing distances from the activation site (at 0 μm; gray dashed line), illustrating electrotonic decrement of propagating voltage signal. $\lambda_{effective}$ (315 μm) was calculated as the distance (black dashed line) at which the recorded potential was 37% of the maximal amplitude at the activation site (purple dashed line). (C) $\lambda_{effective}$ for cables with fixed, narrow, uniform diameter (as in B; $d_0$ = $d_1$=0.5 μm) and varying passive properties. $\lambda_{effective}$ is plotted as a function of axial resistivity ($R_a$ in Ω*cm) for cables with different specific membrane resistivities ($R_m$ in Ω*cm$^2$; plotted in different colors). (D) Top: illustration showing simulated measurement of $\lambda_{effective}$ (as in B, Top) in a cable model with geometry reminiscent of an STG neurite ($d_0$ = 20 μm, $d_1$ = 0.5 μm) and the same passive properties as the cable examined in B and C. Bottom: Plot depicts the amplitude of the evoked inhibitory potential measured at increasing distances from the activation site (as in B, Bottom; $\lambda_{effective}$ >1 mm in this case). (E) $\lambda_{effective}$ for cables with fixed tapering geometry (as in D) and varying passive properties (plotted as in C).

DOI: https://doi.org/10.7554/eLife.41728.002

The following figure supplements are available for figure 1:

**Figure supplement 1.** Simulated measurement of electrotonus in neurites with diverse passive properties and geometries.

DOI: https://doi.org/10.7554/eLife.41728.003

**Figure supplement 2.** Simulated measurement of electrotonus in neurites in cables with unsealed proximal ends.

DOI: https://doi.org/10.7554/eLife.41728.004

junction to ≤2 μm at terminating tips (*Figure 2D–E*, *Figure 2—figure supplement 1–2* to *Figure 2*; *Otopalik et al., 2017a*). Geometrical taper from proximal to distal end is thought to increase the electrotonic length constant for long neurite paths (*Holmes and Rall, 1992*). Yet, we found that, while some STG neurites tapered gradually, others presented diameters that decrease in an abrupt step. Previous study has demonstrated that geometrical taper influences action potential shape and velocity (*Goldstein and Rall, 1974*). But, it is unclear whether this geometrical feature influences the electrotonic decrement of slower inhibitory potentials. Thus, we simulated and measured $\lambda_{effective}$ (as in *Figure 1* and *Figure 1—figure supplement 1–2* to *Figure 1*) in cables with varying degrees of taper from proximal to distal end (*Figure 2—figure supplement 3* to *Figure 2*, Part A). We ran this simulation in cables with varying proximal diameters (ranging between 1 and 10 μm) and maintained the same 80% reduction in diameter from proximal to distal end (*Figure 2—figure supplement 3* to *Figure 2*, Part B). In brief, we found that cables with a gradual taper presented longer $\lambda_{effective}$ than those with abrupt step reductions (consistent with *Holmes and Rall, 1992*). Yet, cables with wide diameters consistent with those measured in STG neurons (*Figure 2E*), presented $\lambda_{effective}$≥ 1 mm for even abrupt step-reductions in diameter for a range of passive properties ($R_a$values ≤ 100 Ω x cm and $R_m$values ≥ 10,000 Ω x cm$^2$). This contrasts with finer cables (tapering from 1 to 2 μm to sub-μm diameters), wherein any step reductions along the path of propagation resulted in $\lambda_{effective}$ < 0.5 mm.

## Measuring electrotonus in four STG neuron types

We experimentally assessed passive voltage signal propagation by evoking inhibitory potentials at numerous sites on the neurite tree with focal photo-uncaging of MNI-glutamate and two-electrode current clamp recordings at the soma (as in *Otopalik et al., 2017b*). *Figure 3A* shows example traces of evoked inhibitory potentials at six sites on an individual PD neurite. When the somatic membrane potential is at rest (approximately −50 mV) evoked events at the six sites are inhibitory but vary in magnitude. Two-electrode current clamp was used to manipulate the somatic membrane potential (between −40 and −100 mV) and the apparent reversal potentials ($E_{rev}$s) for events evoked at each site were determined by plotting response amplitude (measured at the soma) as a function of somatic membrane potential (*Figure 3B*). The x-intercept of the linear fit of these data serves as our measure of the apparent $E_{rev}$ for each site. Across the six sites evaluated in *Figure 3A and B*, apparent $E_{rev}$s ranged between −59 and −70 mV.

Maximal response amplitudes (as measured at a somatic membrane potential of −50 mV) and apparent $E_{rev}$s were measured for numerous sites across the neurite trees of each neuron type (for 10–30 distinct sites across the neurite trees of five LP, VD, GM neurons and six PD neurons; *Tables 1* and *2*). These data are summarized in *Figure 3C and D*, where maximal response amplitudes and apparent $E_{rev}$s for individual sites are plotted as a function of distance from the somatic recording

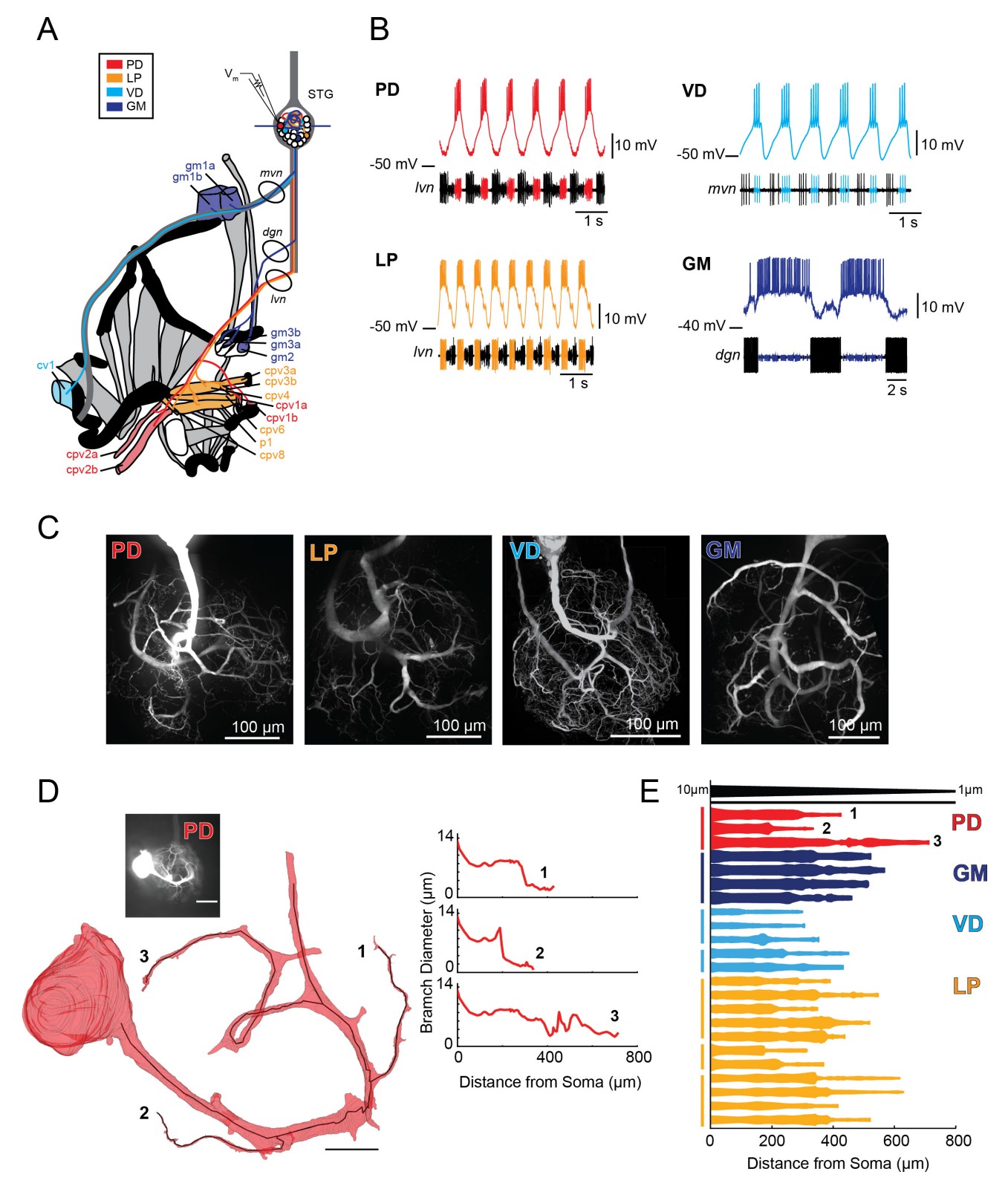

**Figure 2.** Characteristics of four identified STG neuron types. (A) Bilateral innervation patterns of each neuron type, depicted on one side of the dissected foregut. Axons from the four neuron types project from the STG (top) and project to specific muscle groups (indicated with the same colors). The axonal spiking activity for each of these neurons can be recorded with extracellularly nerve recordings at the circled locations on the *mvn, dgn,* and *lvn* nerves. (B) Each neuron type can be identified electrophysiologically by matching intracellular firing patterns with concurrent extracellular recordings

*Figure 2 continued on next page*

*Figure 2 continued*

of nerves known to contain their axons (as shown in A). (**C**) Representative z-projections show complex neurite trees for each neuron type acquired at 40x magnification. (**D**) Left: volumetric reconstruction of soma and a subset of branches in an example PD neuron (inset scale bar 200 μm and black scale ball in reconstruction is 40 μm). Both the full reconstruction (red – used to measure cross-sectional area and infer branch diameter) and skeleton reconstruction (black line – as used to calculate path distance of glutamate photo-uncaging sites to soma in *Figure 3* and Supplements to *Figure 3*) are shown. Right: the diameter of the three branches shown in reconstruction as a function of distance from the soma. (**E**) Flattened representations of the reconstructed branches from a subset of preparations. The width of the shape at a given distance from the soma is directly proportional to the inferred diameter of the branch at that distance. Top: Scaled linearly-tapered branch (black) for reference with a starting width of 10 μm (at x = 0 μm) distal width of 1 μm (at x = 800 μm). For branches in D (right) and E, cross-sectional area and path distance were calculated for each node in the skeleton. Inferred diameters were calculated by treating the cross-section as if it were circular; in actuality very few, if any, cross-sections were exact circles. To reduce abrupt irregularities in the inferred diameter, the plots displayed here are a running average with a sliding window of three skeletal nodes. For A- E: PD = Pyloric Dilator; LP = Lateral Pyloric; VD = Ventricular Dilator; GM = Gastric Mill; *lvn* = lateral ventricular nerve; *dgn* = dorsal gastric nerve; *mvn* = medial ventricular nerve. For all subfigures: red = PD, orange = LP; light blue = VD; dark blue = GM.
DOI: https://doi.org/10.7554/eLife.41728.005

The following figure supplements are available for figure 2:

**Figure supplement 1.** Branch diameter on uncaged branches as a function of path distance from soma.
DOI: https://doi.org/10.7554/eLife.41728.006
**Figure supplement 2.** Branch diameter on uncaged branches as a function of path distance from soma.
DOI: https://doi.org/10.7554/eLife.41728.007
**Figure supplement 3.** Simulated measurement of electrotonus in neurites with varying degrees of taper.
DOI: https://doi.org/10.7554/eLife.41728.008

site for each neuron. For comparison across multiple neurons, maximal response magnitudes (which varied between 0–4 mV) were normalized to the minimum (always 0 mV) and maximum response amplitudes within each neuron (*Figure 3—figure supplement 1–4* to *Figure 3* show raw maximum response amplitudes and apparent $E_{rev}$s for individual PD, LP, VD, and GM neurons, respectively). *Figure 3C* shows that, across all neuron types, maximal response amplitudes show no quantitative trend as a function of distance, nor is there any evidence of normalization of response amplitude with distance (such that response amplitudes are uniform across sites). This is reflected in poorly-fit and insignificant linear regressions of these data (*Table 1*).

*Figure 3D* shows normalized apparent $E_{rev}$s as a function of distance from the somatic recordings site. For comparison of apparent $E_{rev}$s evoked at sites on multiple neurons, apparent $E_{rev}$s were normalized to and plotted as a percent of the mean apparent $E_{rev}$ across sites within each neuron. Horizontal lines denote 0.05, or 5%, above and below the mean $E_{rev}$. For PD, LP, and GM neurons, there appears to be no substantial hyperpolarization in apparent $E_{rev}$s with distance from the somatic recording site (this is validated with linear regressions shown in *Table 1*). *Figure 3B* shows that GM neurons present exceptionally invariant apparent $E_{rev}$s across sites on their neurite trees (mean coefficient of variance (CV) within individual GM neurons was 0.04; *Table 2*). This result is consistent with previous findings (*Otopalik et al., 2017b*). Likewise, LP, and PD neurons exhibit relatively invariant apparent $E_{rev}$s (mean CVs were: 0.06, 0.05, and 0.08, respectively; *Table 2*). It should be noted that a subset of VD neurons shows higher standard deviations and CVs, suggesting heterogeneity of voltage signal propagation across the neuronal structure. Yet, statistical comparison of CVs (ANOVA, [$F_{(3, 17)}=1.2$ p=0.341]) and mean apparent $E_{rev}$s (ANOVA, [$F_{(3, 17)}=2.29$, p=0.1154]) revealed no statistically significant differences across neuron types. This suggests that the four neuron types are similarly electrotonically compact neuronal structures, wherein apparent $E_{rev}$s typically varied by $\leq 10\%$ of the mean within individual neurons. This translates to a range of mean apparent $E_{rev} \pm$ 6–8 mV within each individual neuron.

## Directional sensitivity and voltage integration in diverse neurites

Next, we examined (i) the directional sensitivity of voltage signal propagation and (ii) how multiple voltage events are integrated in single neurites varying in their geometry and passive properties. We first characterized these two properties in our library of cable models with varying geometries and passive properties. *Figure 4* illustrates the measurement of directional sensitivity and summation arithmetic in two cable models: a classic cable model with a narrow diameter (0.5 μm) and a cable

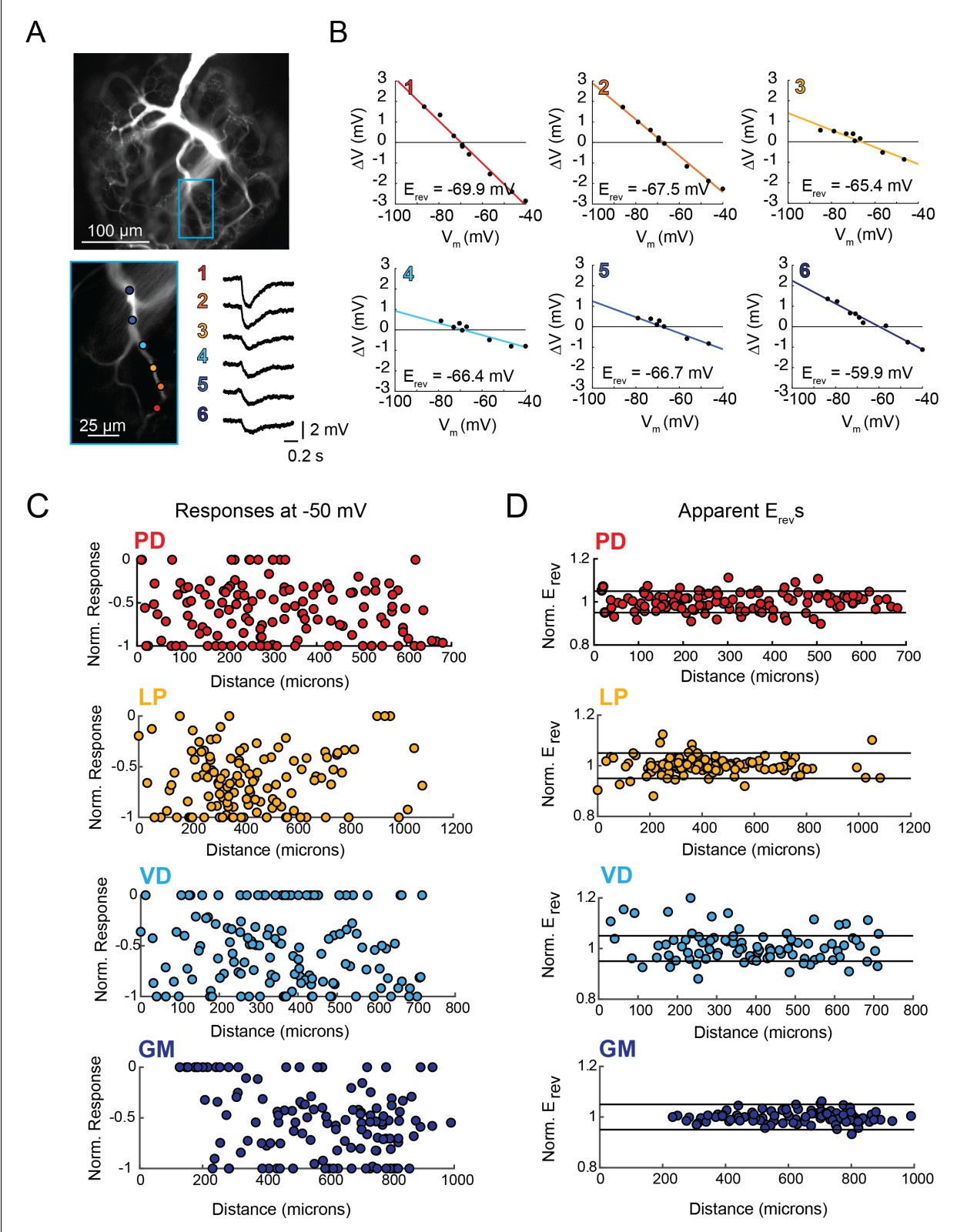

**Figure 3.** Variable response amplitudes and invariant apparent reversal potentials ($E_{rev}$s) across STG neuronal structures. (**A and B**) Glutamate photo-uncaging and measurement of apparent $E_{rev}$s in a representative PD neurite. (**A**) Fluorescence images of an Alexa Fluor 488 dye-fill showing the neurite tree at 20x magnification and a single branch at 40x magnification (outlined in blue on the 20x image). Glutamate photo-uncaging sites are indicated with colored circles and corresponding responses (measured at the soma) for each site are shown. At each site, inhibitory events were evoked with a 1

*Figure 3 continued on next page*

*Figure 3 continued*

ms, 405 nm laser pulse at a starting somatic membrane potential of −50 mV. (B) Plots showing evoked response amplitude (ΔV) as a function of somatic membrane potential ($V_m$). At each site, glutamate responses were evoked at varying somatic membrane potentials (achieved with two-electrode current clamp). These data were fit with a linear regression (colored lines) and the $E_{rev}$ for each site was calculated as the x-intercept of this fit (values for each site shown on bottom right of each plot). C. Response amplitudes plotted as a function of distance from the somatic recording site for each neuron type. Maximum response amplitudes were measured at −50 mV for individual sites and normalized to the maximum response amplitude within each neuron (−1 is equivalent to the maximum response within individual neurons). There was no quantitative relationship between response amplitude and distance (supported by poorly fit linear regression analyses in *Table 1*). D. Apparent $E_{rev}$s for each site were normalized to the mean apparent $E_{rev}$s within each neuron (one is representative to the mean). Horizontal black lines denote boundaries of ±5% of the mean apparent $E_{rev}$s and serve as a graphical depiction of the low variance in apparent $E_{rev}$s within each neuron for sites as far as 800–1000 μm away from the soma. Raw response amplitudes and apparent $E_{rev}$s for individual sites in individual neurons are shown in *Figure 3—figure supplement 1–4* to *Figure 3*.
DOI: https://doi.org/10.7554/eLife.41728.009

The following figure supplements are available for figure 3:

**Figure supplement 1.** PD response magnitudes and apparent reversal potentials ($E_{rev}$s) as a function of distance from the somatic recording site.
DOI: https://doi.org/10.7554/eLife.41728.010
**Figure supplement 2.** LP response magnitudes and apparent reversal potentials ($E_{rev}$s) as a function of distance from the somatic recording site.
DOI: https://doi.org/10.7554/eLife.41728.011
**Figure supplement 3.** VD response magnitudes and apparent reversal potentials ($E_{rev}$s) as a function of distance from the somatic recording site.
DOI: https://doi.org/10.7554/eLife.41728.012
**Figure supplement 4.** GM response magnitudes and apparent reversal potentials ($E_{rev}$s) as a function of distance from the somatic recording site.
DOI: https://doi.org/10.7554/eLife.41728.013

model reminiscent of an STG neurite (with an axial diameter tapering from 20 μm to 0.5 μm; as in *Figure 1*).

Inhibitory events of equivalent conductance magnitude and kinetics (see Materials and methods) were evoked at 500, 600, 700, 800, and 900 μm from the recording site individually and in sequence (5 Hz) in the inward and outward directions (*Figure 4A–D*).

Directional bias was calculated as the integral of the inward response minus the integral of the outward response (*Figure 4E and F*, top). Thus, positive directional biases are indicative of an inward bias in voltage signal propagation (toward the recording electrode) and negative directional biases are indicative of an outward bias in voltage signal propagation. *Figure 4E and F* illustrate a notable difference in the directional sensitivity of passive voltage propagation in these two cable models; the classic cable model shows a robust inward bias and the STG neurite shows no directional sensitivity. These biases persist across a broad range of passive properties. However, the inward bias in the classic cable model does attenuate with increasing axial resistivity ($R_a$) and decreasing membrane resistivity ($R_m$).

The arithmetic of voltage integration was characterized by comparing the combined response to the arithmetic sum of the individual events (example traces depicted in *Figure 4C and D*, bottom). Integration is described as sublinear, linear, or supralinear if the combined response is less than, equal to, or larger than the predicted arithmetic sum. We calculated the arithmetic, or linearity, of the evoked responses as the integral of the inward response minus the integral of the combined individual responses (*Figure 4E and F*, bottom). The classic cable presents sublinear integration of inhibitory voltage events. Yet, as the $R_m$ decreases and $R_a$ increases, voltage summation transitions from sublinear to linear. In contrast, the STG neurite presents relatively linear integration across a broad range of passive properties. Examination of directional sensitivity (*Figure 4—figure supplement 1*) and summation arithmetic in the entire cable library (*Figure 4—figure supplement 2*) demonstrates that a broad range of computations can arise as a consequence of neurite morphology. Taken together, these simulations predict that STG neurites will present directional insensitivity and near-linear voltage integration if they operate predominantly by passive propagation and in the absence of active properties serving to shunt or amplify voltage signals.

## Direction insensitivity in STG neurites

To test for directional bias in voltage signal propagation, sequential voltage events were evoked at multiple sites within the same secondary branches (from tip to primary neurite junction; *Figure 5Ai*). The integrals of the summed responses for inward and outward activation were measured at the

**Table 1.** Linear regression analyses for response amplitudes and apparent reversal potentials (E$_{rev}$s) as a function of distance from the somatic recording site for sites in individual neurons or pooled by cell type.
The data contributing to these analyses are shown graphically in **Figure 3C and D** and **Figure 3—figure supplement 1–4** to **Figure 3**.

| Amplitude vs. Distance | | | | | | Apparent E$_{rev}$ vs. Distance | | | |
|---|---|---|---|---|---|---|---|---|---|
| Neuron | Sites | MSE | R | p | Slope (mV/μm) | MSE | R | p | Slope (mV/μm) |
| PD | 30 | 0.08 | 0.05 | 7.98E-01 | 1.67E-04 | 13.61 | 0.17 | 4.22E-01 | -7.23E-03 |
| PD | 23 | 0.15 | -0.05 | 8.38E-01 | -1.03E-04 | 4.66 | -0.73 | 8.43E-05 | -1.35E-02 |
| PD | 10 | 0.05 | -0.06 | 8.28E-01 | -1.77E-04 | 6.65 | -0.42 | 2.26E-01 | -1.60E-02 |
| PD | 23 | 0.37 | -0.47 | 2.05E-02 | -1.56E-03 | 15.13 | -0.19 | 3.88E-01 | -3.87E-03 |
| PD | 20 | 0.40 | -0.18 | 4.26E-01 | -6.35E-04 | 17.96 | 0.05 | 8.50E-01 | 1.05E-03 |
| PD | 23 | 0.77 | 0.40 | 5.18E-02 | 2.62E-03 | 32.09 | 0.37 | 8.55E-02 | 1.62E-02 |
| PD$_{MEAN}$ | 21.5 | 0.30 | -0.05 | 4.94E-01 | 5.18E-05 | 11.60 | -0.29 | 3.77E-01 | -7.91E-03 |
| PD$_{SD}$ | 6.5 | 0.27 | 0.29 | 3.86E-01 | 1.40E-03 | 9.79 | 0.38 | 3.04E-01 | 1.16E-02 |
| LP | 11 | 0.14 | 0.43 | 1.23E-01 | 5.87E-04 | 12.09 | -0.81 | 2.67E-03 | -1.54E-02 |
| LP | 26 | 0.50 | -0.02 | 9.25E-01 | -1.15E-04 | 5.07 | -0.19 | 3.08E-01 | -4.00E-03 |
| LP | 25 | 0.17 | -0.15 | 4.75E-01 | -5.56E-04 | 14.18 | -0.17 | 4.16E-01 | -5.79E-03 |
| LP | 25 | 0.34 | -0.40 | 4.70E-02 | -1.56E-03 | 5.75 | -0.09 | 6.75E-01 | -2.15E-03 |
| LP | 24 | 0.51 | 0.02 | 9.30E-01 | 7.01E-05 | 0.70 | 0.55 | 5.00E-03 | 7.26E-03 |
| LP$_{MEAN}$ | 22.2 | 0.33 | -0.02 | 5.00E-01 | -3.15E-04 | 10.16 | -0.14 | 2.81E-01 | -4.02E-03 |
| LP$_{SD}$ | 6.3 | 0.18 | 0.30 | 4.22E-01 | 8.08E-04 | 5.45 | 0.48 | 2.86E-01 | 8.12E-03 |
| VD | 27 | 0.31 | 0.05 | 7.95E-01 | 2.32E-04 | 193.71 | -0.35 | 7.64E-02 | -4.17E-02 |
| VD | 19 | 0.54 | -0.53 | 3.49E-03 | -6.51E-03 | 28.63 | -0.45 | 5.57E-02 | -3.85E-02 |
| VD | 24 | 0.28 | -0.14 | 5.06E-01 | -2.46E-04 | 41.44 | 0.22 | 3.16E-01 | 5.07E-03 |
| VD | 15 | 0.07 | -0.20 | 3.36E-01 | -3.28E-04 | 15.74 | -0.29 | 2.97E-01 | -7.76E-03 |
| VD | 15 | 0.87 | 0.35 | 4.73E-02 | 1.81E-03 | 33.58 | -0.69 | 3.29E-03 | -5.59E-02 |
| VD$_{MEAN}$ | 20.0 | 0.42 | -0.09 | 3.38E-01 | -1.01E-03 | 62.62 | -0.31 | 1.50E-01 | -2.78E-02 |
| VD$_{SD}$ | 5.4 | 0.31 | 0.33 | 3.29E-01 | 3.19E-03 | 73.87 | 0.33 | 1.46E-01 | 2.54E-02 |
| GM | 10 | 0.04 | -0.84 | 6.83E-04 | -3.23E-03 | 0.56 | -0.14 | 6.93E-01 | -1.17E-03 |
| GM | 28 | 0.09 | -0.53 | 2.02E-03 | -8.74E-04 | 2.09 | 0.31 | 1.13E-01 | 2.49E-03 |
| GM | 25 | 0.64 | -0.24 | 1.78E-01 | -1.12E-03 | 5.56 | -0.13 | 5.28E-01 | -2.23E-03 |
| GM | 20 | 0.09 | -0.41 | 3.50E-02 | -5.76E-04 | 11.76 | 0.17 | 4.76E-01 | 9.37E-03 |
| GM | 19 | 0.03 | -0.57 | 4.56E-03 | -1.14E-03 | 4.03 | -0.24 | 3.26E-01 | -6.53E-03 |
| GM$_{MEAN}$ | 20.4 | 0.18 | -0.52 | 4.41E-02 | -1.39E-03 | 4.80 | -0.01 | 4.27E-01 | 3.85E-04 |
| GM$_{SD}$ | 6.9 | 0.26 | 0.22 | 7.62E-02 | 1.06E-03 | 4.33 | 0.23 | 2.19E-01 | 5.97E-03 |

DOI: https://doi.org/10.7554/eLife.41728.014

soma (**Figure 5Aii**). Directional preference for each branch was assessed by plotting the response integrals for the inward and outward directions against each other and comparison with the identity line, which is indicative of direction insensitivity wherein the inward and outward response integrals are equal (**Figure 5Aiii**). Any branches with points left of the identity line present an inward, or centripetal bias, whereas any points right of the identity line are suggestive of a centrifugal, or outward, bias. Interestingly, all four neuron types show little directional selectivity (this is supported by rootmean-square error values (RMSE) <0.5 mV*s, a measure of goodness-of-fit to the identity line). Example traces and direction selectivity plots for each cell type can be found in **Figure 5—figure supplement 1–4** to **Figure 5**. Taken together, these results suggest that neurites in each of these four STG neuron types do not exhibit the directional selectivity that has been described in other

**Table 2.** Apparent Mean $E_{rev}$s, standard deviations (SD) and coefficients of variance (CV) for individual neurons and within neuron type.

| Neuron | Sites | Branches | Mean Erev (mV) | SD (mV) | CV |
|---|---|---|---|---|---|
| PD | 30 | 6 | -64.70 | 3.80 | 0.06 |
| PD | 23 | 4 | -66.30 | 3.20 | 0.05 |
| PD | 10 | 3 | -74.30 | 3.00 | 0.04 |
| PD | 23 | 4 | -64.18 | 4.05 | 0.06 |
| PD | 20 | 4 | -69.56 | 4.35 | 0.06 |
| PD | 23 | 4 | -83.00 | 6.22 | 0.08 |
| PD$_{MEAN}$ | 21.5 | 4.2 | -70.34 | 4.10 | 0.06 |
| LP | 11 | 2 | -65.72 | 6.18 | 0.09 |
| LP | 26 | 5 | -72.82 | 2.33 | 0.03 |
| LP | 25 | 5 | -71.87 | 3.89 | 0.05 |
| LP | 25 | 5 | -81.67 | 4.07 | 0.05 |
| LP | 24 | 5 | -71.87 | 3.89 | 0.05 |
| LP$_{MEAN}$ | 22.2 | 4.4 | -72.79 | 4.07 | 0.05 |
| VD | 27 | 4 | -76.22 | 3.75 | 0.05 |
| VD | 19 | 4 | -80.83 | 6.14 | 0.08 |
| VD | 24 | 4 | -83.86 | 6.75 | 0.08 |
| VD | 15 | 4 | -74.81 | 4.29 | 0.06 |
| VD | 15 | 5 | -77.84 | 8.23 | 0.11 |
| VD$_{MEAN}$ | 20 | 4.2 | -78.71 | 5.83 | 0.08 |
| GM | 10 | 2 | -62.33 | 0.80 | 0.01 |
| GM | 28 | 5 | -74.00 | 1.55 | 0.02 |
| GM | 25 | 6 | -74.78 | 2.43 | 0.03 |
| GM | 20 | 5 | -75.42 | 3.57 | 0.05 |
| GM | 19 | 4 | -65.66 | 7.05 | 0.11 |
| GM$_{MEAN}$ | 20.4 | 4.4 | -70.44 | 3.08 | 0.04 |

DOI: https://doi.org/10.7554/eLife.41728.015

neuron types (*Barlow and Levick, 1965*; *Euler et al., 2002*; *London and Häusser, 2005*; *Branco et al., 2010*).

## Arithmetic of voltage signal integration

If voltage signal propagation in STG neurons is predominantly shaped by passive properties and tapered neurite geometries, and less so by active biophysical properties, which may shunt or amplify propagating voltage signals, we would expect to observe linear voltage summation as predicted by our simulation (*Figure 4* and *Figure 4—figure supplement 2*). To assess the arithmetic of voltage summation, we calculated the arithmetic sum of responses evoked at individual sites across single neurites (*Figure 6Ai–iii*; offset by 200 ms to mimic a 5 Hz sequential activation rate). The integrals of the measured response and expected arithmetic sum for a given branch were plotted against each other and compared with the identity line, which is indicative of linear summation, wherein the measured and expected response integrals are equal (*Figure 6Aiii*). Any branches with points left of the identity line present sublinear summation, whereas any points right of the identity line are suggestive of supralinear summation. *Figure 6B* shows these plots for more than 20 branches for each neuron type. Across all neuron types, the majority of branches showed linear summation. RMSE values were less than 1.5 mV*s; thus, the measured response integrals were within 1.5 mV*s of the integral expected of linear summation. GM neurons (*Figure 6Biv*) exhibited particularly uniform linear summation across all branches evaluated and this is reflected in a small RMSE value of 0.32 mV*s. LP branches and PD branches show slightly higher RMSE values (greater than 1 mV*s), perhaps

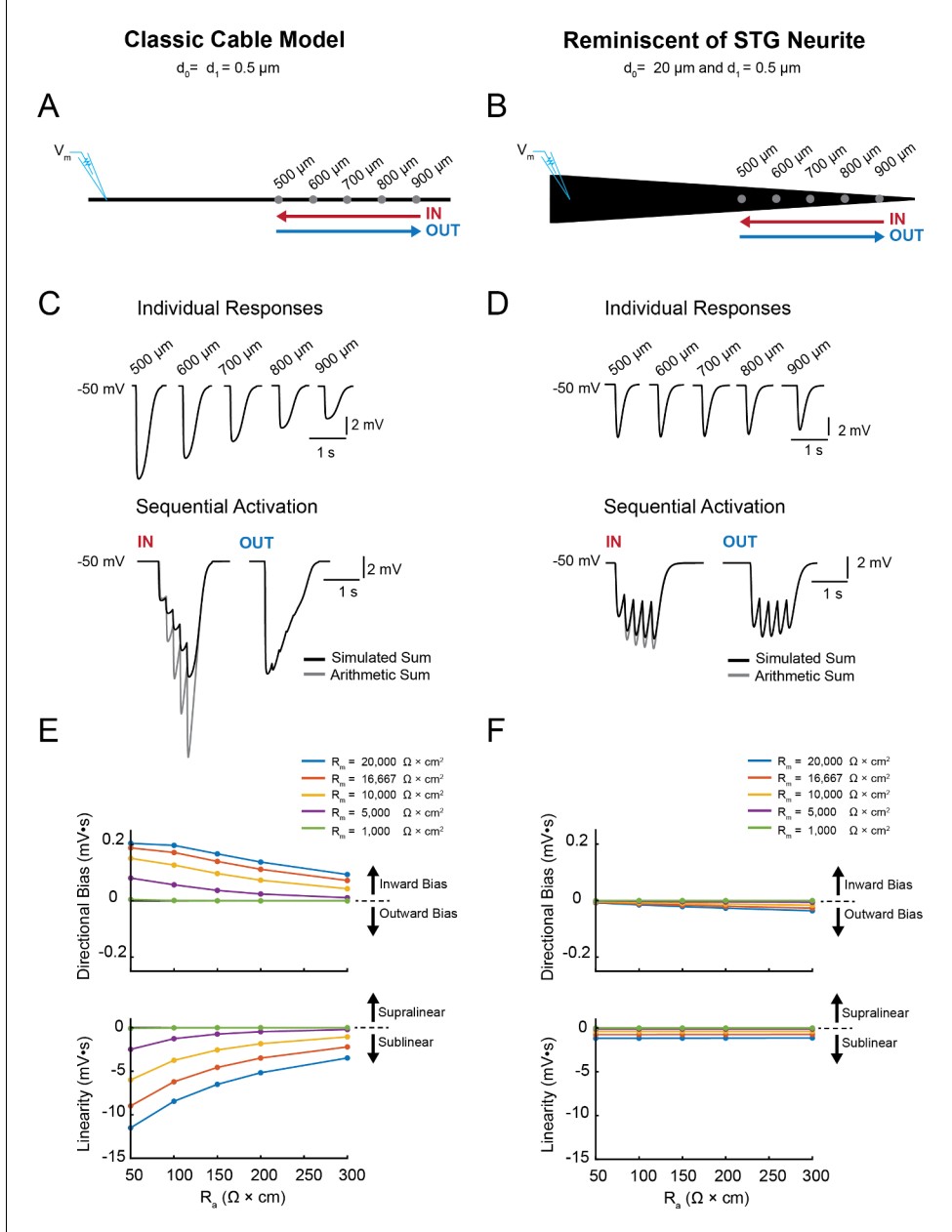

**Figure 4.** Simulating voltage summation in neurites with diverse passive properties and geometries. Voltage summation experiments were simulated in NEURON. Inhibitory potentials were evoked at five sites (500, 600, 700, 800, and 900 μm away from the recording electrode) individually or in sequence at 5 Hz in the inward or outward directions relative to the recording site. (**A-C**) summarizes results from a classic cable model without a tapering geometry, as in **Figure 1B**. (**D-F**) summarizes results from a cable model with neurite geometry reminiscent of that observed in STG neurons, as in **Figure 1D**. (**A**) Schematic showing a classic cable model simulation for a 0.5 μm, uniform-diameter cable. (**B**) Simulated traces show consequence of activating individual sites (top) and sequential activation in either direction (below), as depicted by colored arrows in (**A**). (**C**) Quantification of directional bias and linearity for 0.5 μm-diameter cables varying in their passive properties. Top: directional bias as a function of specific axial resistivity ($R_a$ in Ω*cm). Directional bias was calculated as inward integral minus the outward integral; positive values suggest an inward bias, whereas negative values suggest an outward bias. Points close to the y = 0 suggest no directional preference. Bottom: Linearity as a function of $R_a$. Linearity was calculated as the inward integral minus the integral of the arithmetic sum of events evoked at individual sites (as in B, top); positive values suggest supralinear summation, whereas negative values suggest sublinear voltage summation. Points close to the y = 0 suggest linear summation. All plots show data for a range of specific membrane resistivity ($R_m$ in Ω*cm$^2$)
*Figure 4 continued on next page*

*Figure 4 continued*

values in different colors (indicated in the key below). Data for the full simulation exploring a broad parameter space are shown in *Figure 4—figure supplement 1* and *2* to *Figure 4*. (D) Schematic showing a cable model simulation for a neurite that tapers from 20 μm at the recording end ($d_0$) to 0.5 μm at the distal end ($d_1$). (E) Simulated traces for activation of individual sites (top) and sequential activation in either direction (below), as depicted by colored arrows in D. (F) Directional bias and linearity plotted as a function of $R_a$ and varying $R_m$ values (indicated in key) for a cable with the geometry shown in D. B and E: $R_m$ = 10000 Ω*cm$^2$ and $R_a$ = 100 Ω*cm.

DOI: https://doi.org/10.7554/eLife.41728.016

The following figure supplements are available for figure 4:

**Figure supplement 1.** Simulated measurement of directional bias of voltage summation in neurites with diverse passive properties and geometries.

DOI: https://doi.org/10.7554/eLife.41728.017

**Figure supplement 2.** Simulated measurement of voltage summation arithmetic in neurites with diverse passive properties and geometries.

DOI: https://doi.org/10.7554/eLife.41728.018

providing some evidence of variable shunting or amplifying mechanisms. Even so, RMSE values less than 2 mV*s across all cell types suggests relatively linear summation across most branches evaluated. Voltage summation arithmetic was similarly linear in the centrifugal direction (consult *Figure 5—figure supplement 1–4* to *Figure 5*). It should be noted that these experiments were performed in the absence of TTX-sensitive sodium channels, which have been abolished with $10^{-7}$ M TTX in the bath.

## Discussion

### Compact computing in STG neurons

In the present study, we find that multiple STG neuron types present electrotonically compact structures and within-neurite voltage summation that is relatively linear and directionally insensitive. Taken together, these findings suggest that STG neurons are built to linearly sum and unify synaptic inputs distributed across their expansive and complex structures, rather than perform distributed computations on a branch-by-branch, or subtree-by-subtree basis. Our computational simulations suggest that this biophysical architecture may be achieved passively and as a simple consequence of neurite geometry in these neurons, which present neurites with wide diameters that taper from 10 to 20 μm near the soma, to sub-micron diameters at their terminating tips.

This computing strategy stands in contrast to that which has been observed in other neuron types with similarly complex neuronal structures. When voltage signals propagate long distances in the absence of amplifying mechanisms, electrotonic decrement is thought to result in sublinear voltage integration and directional bias within single branches (*Rall, 1964*; *Gulledge et al., 2005*; *London and Häusser, 2005*). This has been demonstrated experimentally in a variety of vertebrate neuron types in different circuit contexts, from cortex to retina (*Barlow and Levick, 1965*; *Cash and Yuste, 1998*; *Euler et al., 2002*; *Poirazi et al., 2003*; *Polsky et al., 2004*; *Branco et al., 2010*). Linear and supralinear integration are thought to require amplifying mechanisms, such as voltage-gated ion channels or distance-dependent scaling of receptors (*Magee and Cook, 2000*; *Andrasfalvy and Magee, 2001*; *Smith et al., 2003*; *Gulledge et al., 2005*; *Lavzin et al., 2012*). Collectively, these studies support the notion that highly-branched and expansive neuronal structures, are likely to present some degree of passive voltage attenuation (in the absence of amplifying mechanisms) and perform compartmentalized computations as a consequence. Yet, this framework for dendritic computation does not rely on the study of, or account for, the diverse dendrite and neurite morphologies observed in nature.

To make sense of our findings in the STG, we revisited basic cable theory and simulated electrotonus and passive voltage integration in a library of neurites exhibiting diverse geometries and passive biophysical properties. We observed a surprising range of electrotonic decrement, directional bias, and voltage summation arithmetic across the surveyed morphological and biophysical parameter space. Thus, this simulation revealed multiple morphological and biophysical solutions for

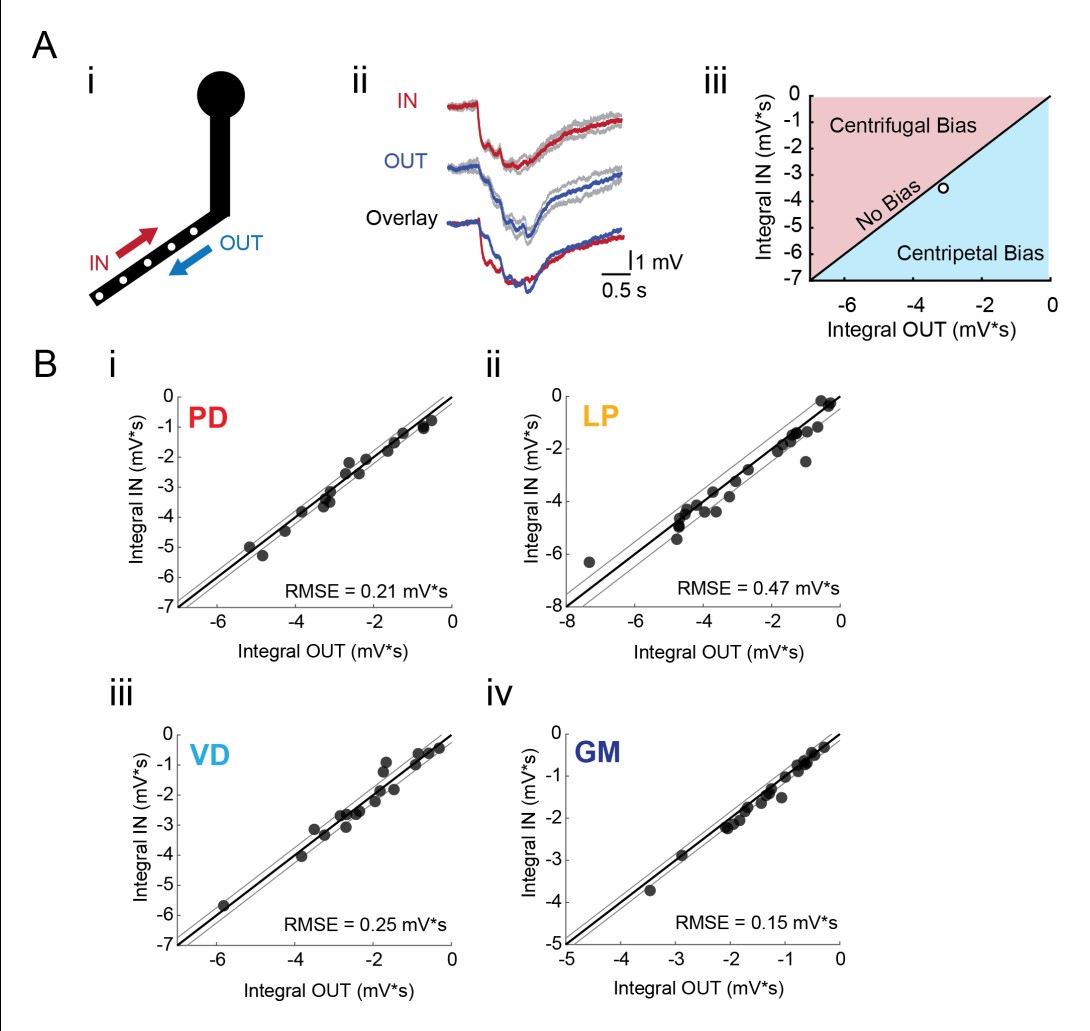

**Figure 5.** Directional sensitivity of voltage propagation in four STG neuron types. (**A**) (i) 4–6 sites on single secondary neurites were sequentially photo-activated at 5 Hz in the inward (IN) or outward (OUT) directions. The integrals (ii) of these inhibitory summation responses were calculated as the area above the trace (in mV*s) and plotted against each other as shown in (iii). As plotted, any points that lie to the right of the identity line (shaded in blue) show a centripetal, or inward bias, whereas any points that lie to the left of the identity line (shaded in red) show an outward, or centrifugal, bias. Any points near or on the identity line are unbiased (as is the case with the example traces shown in (ii), depicted with the white data point in (iii)). (**B**) Directional bias plots for numerous branches within neuron type: (i) 19 branches from 6 PD neurons, (ii) 27 branches from 9 LP neurons, (iii) 20 branches from 5 VD neurons, (iv) 22 branches from five neurons. These data were fit to the identity line, and the root-mean-square error (RMSE) boundaries for this fit is plotted in gray lines.

DOI: https://doi.org/10.7554/eLife.41728.019

The following figure supplements are available for figure 5:

**Figure supplement 1.** Directional sensitivity and arithmetic of voltage propagation in PD neurons.

DOI: https://doi.org/10.7554/eLife.41728.020

**Figure supplement 2.** Directional sensitivity and arithmetic of voltage propagation in LP neurons.

DOI: https://doi.org/10.7554/eLife.41728.021

**Figure supplement 3.** Directional sensitivity and arithmetic of voltage propagation in VD neurons.

DOI: https://doi.org/10.7554/eLife.41728.022

**Figure supplement 4.** Directional sensitivity and arithmetic of voltage propagation in GM neurons.

DOI: https://doi.org/10.7554/eLife.41728.023

achieving varying degrees of passive electrotonic decrement and different computational strategies. We also show that linear summation and directional insensitivity, as observed in four STG neuron types, can be achieved in the absence of active properties altogether and in the face of potential heterogeneity of passive biophysical properties. Altogether, this proof-of-concept simulation both recapitulates the principles describe in early theoretical studies and also demonstrates that different biophysical and morphological strategies are likely to be utilized by different neuron types, to suit their unique physiological function. Thus, the widely-accept principles derived from the study of so-called canonical neuron types may be less general than previously thought.

## A general solution for reliable pacemaking physiology

Our findings are interesting in light of the synaptic organization of the STG. Synaptic sites are sparsely distributed throughout the neuropil and pre- and post-synaptic sites are closely apposed on the same neurites (*King, 1976a*; *King, 1976b*; *Kilman and Marder, 1996*). Thus, sufficiently large synaptic potentials may originate anywhere on the neurite tree, propagate in any direction, and achieve consistent neuronal output.

The motor rhythm generated by the STG relies on slow oscillations and graded inhibitory transmission (*Eisen and Marder, 1982*; *Marder and Eisen, 1984*; *Maynard and Walton, 1975*; *Graubard et al., 1980*; *Manor et al., 1997*; *Manor et al., 1999*; *Bose et al., 2014*; *Golowasch et al., 2017*). The biophysical features we have described in STG neurons may allow for the averaging of very large and slow synaptic conductances in space and time, thereby sustaining pattern generation in this motor circuit. Rhythmic networks that compute with slow oscillations and/or graded transmission (*Walsh et al., 1972*; *Wilson and Wachtel, 1974*; *Pearson and Fourtner, 1975*; *Robertson and Pearson, 1985*; *Angstadt and Calabrese, 1991*; *Dicaprio, 1989*; *Dicaprio et al., 1997*; *Dale, 1995*; *DiCaprio, 2003*; *Smarandache-Wellmann et al., 2013*) may benefit from a similar biophysical and morphological architecture. Moreover, we suggest that wide neurites may aid STG neurons in generating consistent physiological output, in the face of variable conductance magnitudes and subcellular distributions of intrinsic and synaptic properties across animals (*Prinz et al., 2004*; *Schulz et al., 2006*; *Marder and Goaillard, 2006*; *Goaillard et al., 2009*; *Marder, 2011*).

In the present study, we assessed the integration of inhibitory glutamate events in the presence of TTX, which blocks voltage-gated sodium currents and silences circuit activity. Thus, we can only speculate how action potentials, arising from such TTX-sensitive currents, may influence the integration of graded inhibitory glutamatergic synaptic events in the intact circuit. We would not expect TTX-sensitive currents to substantially alter inhibitory voltage signal propagation or integration at the range of membrane potentials probed here ($-100$ to $-40$ mV for apparent reversal potentials measurements and $-50$ mV for the voltage summation and directional sensitivity measurements). Moreover, spike initiations zones, where such TTX-sensitive channels are most likely to reside, are located just outside the neuropil, where the axons exit the ganglion (*Raper, 1979*; *Miller, 1980*). PD neurons exhibit a second, dopamine-sensitive axonal spike initiation zone between the upper *dvn* and its split into the bilateral *lvns* (refer to *Figure 2A*; *Bucher et al., 2003*). Given the peripheral locations of the spike initiation zones in these neurons, it is unlikely that TTX-sensitive voltage-gated channels would shunt the current arising from these evoked events in the same way as has been seen in other systems (*Laurent, 1990*). Spatial separation of spike initiation zones from synaptic integration and slow wave generation in the neuropil may reduce shunting of synaptic currents. A thorough investigation of this possibility requires experiments in varying pharmacological and modulatory conditions, which may reveal how various currents and ongoing circuit activity may influence these voltage propagation and integration in these otherwise compact passive neuronal structures.

Taken together with previous work (*Otopalik et al., 2017a*; *Otopalik et al., 2017b*), the present study suggests that, given the relatively compact electrotonic architecture of STG neurons arising from their neurite geometries, other features of their morphologies and the exact spatial organization of synaptic contacts may not be critical determinants of the distinct physiological waveforms and firing patterns exhibited by different STG neuron types. Thus, one is left with the expectation that their different activity patterns are predominantly determined by their cell-type-specific ion channel and receptor expression profiles. And, indeed, there is much evidence that the 14 different neuron types express different palettes of receptors (*Swensen et al., 2000*; *Swensen and Marder,*

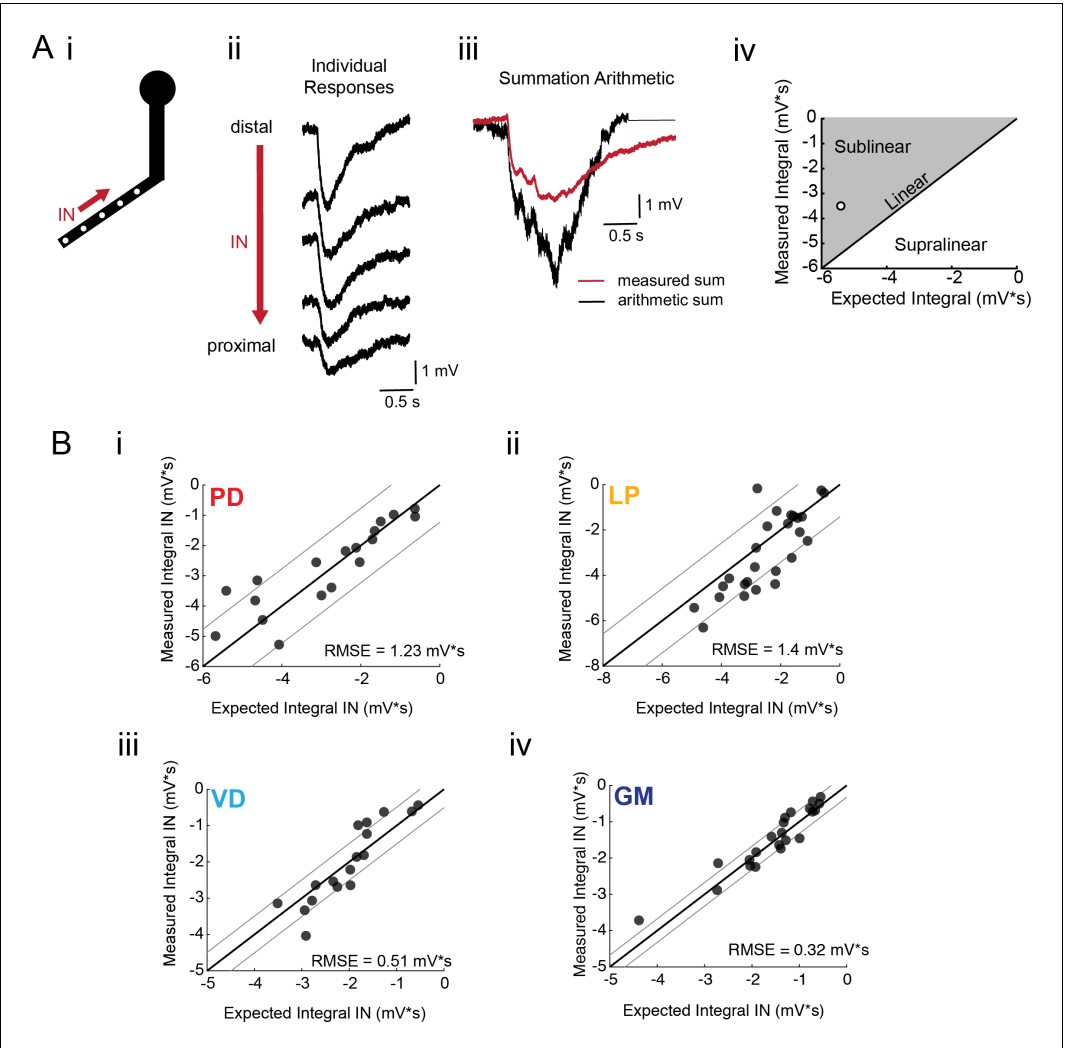

**Figure 6.** Arithmetic of voltage propagation in four neuron types. (**A**) (**i**) 4–6 sites spaced 50–100 μm apart on the same secondary neurite were sequentially photo-activated at 5 Hz in the inward (IN) direction. (**ii**) Raw responses at individual sites from most distal (top) to most proximal (bottom) for one representative PD branch. (**iii**) Comparison of the inward voltage sum to the arithmetic sum for photo-activation of the individual sites (the traces shown in (**ii**) were summed with a 200 ms offset). (**iv**) The integrals (mV*s) of the measured responses were plotted against that of the expected arithmetic sum. This provides a graphical depiction of the linearity of voltage summation for each branch. Points to the left of the identity line suggest sublinear summation, points to the right of the identity line suggest supralinear summation, and points near or on the identity line suggest linear summation. The singular point depicted in (**iv**) depicts voltage summation for the responses shown in (**iii**). In this case, the measured voltage had a lesser integral than the arithmetic sum and therefore showed sublinear summation. (**B**) (**i–iv**) Plots showing the measured integrals as a function of the expected integral for the arithmetic sum for the inward activation of many branches within the four neuron types: (**i**) 19 branches from 6 PD neurons, (**ii**) 27 branches from 9 LP neurons, (**iii**) 20 branches from 5 VD neurons, (**iv**) 22 branches from five neurons. These real data were fit to the identity line, and the root-mean-square error (RMSE) boundaries for this fit is plotted in gray lines.

DOI: https://doi.org/10.7554/eLife.41728.024

2000; *Swensen and Marder, 2001*) and cell-type-specific ratios of ion channels (*Schulz et al., 2006*; *Schulz et al., 2007*).

Of course, one is left wondering why these neurons present such expansive and complex morphologies, if they instead act as single electrotonic compartments. One possibility is that this allows STG neurons to make appropriate synaptic contacts wherever partner neurons ramifying throughout the neuropil find each other. Additionally, it is likely that STG neurons grow to fill distinct spatial

fields in the neuropil (*Otopalik et al., 2017a*) for the purpose of maximizing surface area for the reception of the many diffuse neuromodulatory substances that are released in the hemolymph and by descending modulatory inputs of the stomatogastric nerve (*Marder and Bucher, 2007*; *Blitz and Nusbaum, 2011*).

## Materials and methods

### Animals and dissections
Wild-caught adult male Jonah Crabs (*Cancer borealis)* were acquired and maintained by the Marine Resources Center at the Marine Biological Laboratories in Woods Hole, MA. Animals were maintained on a 12 hr dark/12 hr light cycle without food and in chilled natural seawater (10–13 deg C) in a 2000-liter tank at a density of no more than 30 crabs per tank. STG dissections were executed as in *Otopalik et al. (2017b)* and as previously described (*Gutierrez and Grashow, 2009*) in saline solution (440 mM NaCl, 11 mM KCl, 26 mM MgCl$_2$, 13 mM CaCl$_2$, 11 mM Trizma base, 5 mM maleic acid, pH 7.4–7.6). The intact stomatogastric nervous system, including: two bilateral commissural ganglia, esophageal ganglion, and stomatogastric ganglion (STG), as well as the *lvn*, *mvn*, *dgn* were dissected from the animal's foregut and pinned down in a Sylgard-coated petri dish (10 mL). The preparation was continuously superfused with chilled saline (11–13 degrees C) for the duration of the experiment using a bipolar temperature control system (Harvard Apparatus, CL-100).

### Electrophysiology and Dye-fills
All electrophysiology and dye-fill methods are consistent with those utilized in *Otopalik et al. (2017b)*. The STG was desheathed for access to somata for intracellular recordings. These recordings were executed with glass micropipettes (20–30 MΩ) filled with internal solution: 10 mM MgCl$_2$, 400 mM potassium gluconate, 10 mM HEPES buffer, 15 mM NaSO$_4$, 20 mM NaCl (*Hooper et al., 2015*). Intracellular recordings signals were amplified with an Axoclamp 900A amplifier (Molecular Devices, as described in *Otopalik et al. (2017b)*. For extracellular nerve recordings, Vaseline wells were built around the *lvn, mvn,* and *dgn* nerves and stainless-steel pin electrodes were used to monitor extracellular nerve activity (as indicated in *Figure 1A*). Extracellular nerve recordings were amplified using a Model 3500 extracellular amplifier (A-M Systems). All recordings were acquired with a Digidata 1550 (Molecular Devices) digitizer and visualized with pClamp data acquisition software (Axon Instruments, version 10.7). Neuron types were identified by matching concurrent intracellular spiking patterns with units on nerves known to contain their axons (as in *Figure 1B*) and verified with positive and negative current injections. After identification, a single neuron was filled with dilute alexa488 dye (2 mM Alexa Fluor 488-hyrazide sodium salt (ThermoFisher Scientific, catalog no. A-10436, dissolved in internal solution)) with negative current pulses (−4 nA, 500 ms at 0.5 Hz) for 15–25 min. Following the dye-fill, input resistance was measured at the soma in two-electrode current clamp (neurons with input resistances < 5 MΩ were discarded). For two-electrode current clamp, the electrode containing dilute alexa488 was used for recording and amplified on a 0.1xHS headstage. The electrode used for cell identification was used for current injection and amplified with a 1xHS headstage. Input resistance was measured throughout the experiment and neurons with input resistances < 5 MΩ were discarded. Reversal potentials for glutamate-evoked responses were determined by evoking responses at ≥eight membrane potentials between −100 and −40 mV. In some experiments, neurons were filled with 2% Lucifer Yellow CH dipotassium salt (LY; Sigma, catalog no. L0144; diluted in filtered water) for post-hoc imaging. LY was injected with a low-resistance (10–15 MΩ) glass micropipette for 20–50 min with negative current pulses (−six to −8 nA, 500 ms at 0.5 Hz).

### Focal glutamate Photo-uncaging
Focal glutamate photo-uncaging methodology was consistent with the methods used in *Otopalik et al. (2017b)*, although different instrumentation was used. For photo-uncaging experiments, preparations were superfused with a re-circulating peristaltic pump to maintain a stable bath volume. 250 μM MNI-caged-L-glutamate (dissolved in saline; Tocris Bioscience, catalog no. 1490) was bath applied. $10^{-7}$ M teterodotoxin (TTX) was also superfused to minimize spike-driven synaptic activity. Alexa Fluor 488-filled neurons were visualized and focal photo-uncaging was achieved with

a Laser Applied Stimulation and Uncaging (LASU) system (Scientifica). In brief: this system was composed of an epifluorescence microscope (SliceScope, Scientifica) equipped with a 4x magnification air and 40x magnification water-immersion objective lenses (Olympus; PLN 4X and LUMPLFLN 40XW, respectively). A 780 IR-LED was used to visualize the stomatogastric ganglion and locate neurons of interest. A white fluorescence illumination system (CoolLED) and FITC/Alexa Fluor 488/Fluo3/Oregon Green filter set (Chroma) were used to excite and visualize fluorescent emission from neuronal dye-fills. Images were captured with a monochrome CCD camera (Scientifica, SciCam Pro; $1360 \times 1025$ array and 6.54 $\mu m^2$ pixel size). Focal photo-activation of MNI-glutamate was achieved with a 405 nm laser (1 ms pulses, 35 mW, spot size $\leq$1.5 $\mu$m with the 40x objective). The preparation platform and micromanipulators were mounted on a motorized movable base plate, allowing for smooth re-positioning (in the X-Y plane) of the objective over different neurites. For photo-activation at multiple sites within the field of view, the laser spot was re-positioned in quick sequence (5 Hz) using a set of X-Y galvanometers (Cambridge Technology, 6251H). Photo-uncaging sites within the field of view, laser pulse duration, and pulse rate were selected with the assistance of the LASU system software (Scientifica).

## Electrophysiology analysis

Electrophysiological responses to focal glutamate photo-uncaging were visualized and analyzed offline, as previously described (*Otopalik et al., 2017b*), using a set of custom MATLAB (Mathworks, version 2017b) scripts that will be made available on the Marder Lab GitHub (https://github.com/marderlab) (*Otopalik, 2019*; copy archived at https://github.com/elifesciences-publications/Otopalik-Pipkin-Marder-2019). Maximal response amplitudes, directional bias, and voltage integration arithmetic were calculated as an average of 3 trials of photo-stimulation at individual sites or of individual branches, accordingly. Apparent reversal potentials ($E_{rev}$s) were calculated for individual sites by plotting response amplitudes as a function of somatic membrane potential and fitting these scatter plots with a linear regression. The apparent $E_{rev}$ for each site was calculated as the x-intercept of this fit. These methods are described in detail in *Otopalik et al. (2017b)*. The For the linear regressions for all sites depicted in *Figure 3D* and supplements (plots in Part C), R-values were >0.85.

## Post-hoc imaging and morphological analysis

For each neuron, all photo-uncaging sites were re-located in fluorescence images of the Alexa Fluor 488 and/or Lucifer Yellow dye-fills acquired with the LASU microscope (at 40x and 20x magnification; described above). The distance between each site and the somatic recording site was measured utilizing a combination of 2-D and 3-D image stacks with the assistance of Simple Neurite Tracer on ImageJ/FIJI (*Longair et al., 2011*). Although most experiments were exclusively conducted using the LASU microscope, a subset of neurons were imaged at 40x magnification (Zeiss C-apochromat40x/1.2 W) on a VIVO microscope system equipped with a Yokogawa (CSU-X1, Japan) spinning disk confocal scan head mounted on a Zeiss Examiner microscope. Fluorescent dye-fills were visualized with standard GFP filters and images were captured with a Prime CMOS camera (Photometrics, 95B). Multiple image stacks in the z-dimension, spanning the STG, were stitched together with the assistance of the Stitching tool in ImageJ/FIJI (*Preibisch et al., 2009*). Maximum projections of these stacks are shown in *Figure 2C*.

## Volumetric neurite reconstructions

For high-resolution imaging and volumetric neurite reconstructions, a subset of preparations were fixed in 4% paraformaldehyde following photo-uncaging experiments and kept at 4° C until immunostaining. For immuno-staining, each preparation was incubated with a rabbit anti-Lucifer Yellow IgG antibody (Invitrogen A5750) overnight at room temperature in phosphate-buffered saline (PBS) containing 0.1% Tris, then washed and subsequently incubated with a goat anti-rabbit IgG secondary antibody conjugated to Alexa-488 (Invitrogen A11034) in PBS at room temperature. Preparations were mounted on slides using VectaShield (Vector Laboratories H-1000). Seven preparations survived this full sequence, from electrophysiology and glutamate photo-uncaging, through immuno-staining, mounting, and imaging: one PD, three LPs, two VDs, and one GM.

High-resolution confocal images of each preparation were taken on a Leica SP5 system. We used a 63x objective to collect a montage of Z-stacks spanning the entire neuropilar arbor of the labeled

cell at a voxel resolution of 0.114 µm x 0.114 µm in the X-Y plane and 0.797 µm in Z dimension. We found this resolution was sufficient to relocate and trace the branches that were subjected to glutamate photo-uncaging. Image montages were aligned and reconstructions acquired within the TrakEM2 (fiji.sc/TrakEM2, RRID: SCR_008954; *Cardona et al., 2012*) environment of ImageJ/FIJI. We fully traced each neuron from the origin of the primary neurite near the soma to the tips of each branch. As our analyses concerned only the branches visited for photo-uncaging, our reconstructions therefore do not recapitulate the entirety of the neuronal arbor. We traced each preparation in two ways. First, we generated a skeleton wherein the neuron is represented by a tree of connected nodes. Second, we traced the full volume of each branch containing the skeleton by manually coloring in a region in each z-layer that contained that branch.

Using these two reconstructions for each neurite, we measured the path distance of each node in the skeleton to the soma along with the cross-sectional area of the branch at each node. From these cross-sectional areas, we calculate an approximated diameter of the branch assuming the branch were cylindrical; in most cases, of course, the branches are not perfectly cylindrical. To generate cross-sectional area, we used the built-in coding environment of FIJI to write a custom Python script. Briefly, the algorithm generates a list of all points on the edge of branches in the arbor, then finds those which lie within a small threshold (0.5 µm) of a plane orthogonal to the arbor of the branch at each node. The number of these points are further reduced by excluding those which lie outside a certain distance of the node (which corresponds to the maximum expected radius of the branch at that node). For most branches, we used a distance threshold of 3 µm, while for thicker branches we used 11 µm. Finally, the projection of these points onto the orthogonal plane through the branch at each node defines a polygon of which the area is calculated using the shoelace formula.

## Passive cable models

The library of cable models utilized in *Otopalik et al. (2017b)* was adapted to explore electrotonus, directional bias, and voltage integration in neurites with varying geometries (depicted in *Figure 6*) and passive properties: six specific axial resistances ($R_a$): 10, 50, 100, 150, 200, 300 Ω*cm; and six specific membrane resistances ($R_m$): $2 \times 10^4$, $1.6 \times 10^4$, $1 \times 10^4$, $5 \times 10^3$, $1 \times 10^3$, $1 \times 10^2$ Ω*cm². All cables were 1000 µm in length and had a membrane capacitance of 1 µF*cm$^{-2}$. Electrotonus was assessed as described in *Figure 1* and voltage summation was assessed by simulating our experimental glutamate photo-uncaging procedure using the simulation platform NEURON (*Hines and Carnevale, 2001*). All possible combinations of neurite geometries, membrane resistances, and axial resistances were assessed in *Figure 1—figure supplement 1* to *Figure 1*, *Figure 4*, and *Figure 4—figure supplement 1* and *2* to *Figure 4*, whereas a representative cross-section of cable models are presented in *Figure 1—figure supplement 2* to *Figure 1* and *Figure 2—figure supplement 3* to *Figure 2*. To assess voltage integration directional bias and arithmetic, voltage was recorded 100 µm from the proximal end ($d_0$) and inhibitory potentials ($E_{rev}$ = −75 mV, τ = 70 ms, $g_{max}$ = 10 nS) were evoked at five sites with increasing distance from the recording site (as depicted in *Figure 4A, B*). Sites were activated individually, and then at 5 Hz in the inward (toward the recording site) or outward (away from the recording site) directions. Using MATLAB (Mathworks, version 2018b), integrals were calculated for the inward and outward summed responses and the expected linear sum of the individual events in either direction. Directional bias was calculated as the inward integral minus the outward integral. Linearity was calculated as the expected arithmetic sum minus the recorded voltages sum for the inward direction. Custom scripts written to generate cable models and simulate experimental procedures in NEURON were composed in Sublime Text (Sublime HQ Pty Ltd, Sydney) and can be found on the Marder Lab GitHub (https://github.com/marderlab), along with all simulation output.

## Acknowledgements

We thank Jennifer Bestman for assistance in spinning disk and confocal microscopy; the Marine Resources Center at the Marine Biological Laboratories for acquiring and maintaining animals; Louie Kerr at the Central Microscopy Facility; Dana Mock-Munoz de Luna for administrative support; Kamran Kodhakah, Heather Rhodes, and the 2017 Grass Fellows for their support and feedback; and lastly, Edward Dougherty at the Brandeis University Confocal Imaging Lab for support and

microscope maintenance. This study was funded by the Grass Foundation and NINDS awards to F31NS092126 to AO and R35NS097343 to EM.

## Additional information

### Competing interests
Eve Marder: Deputy Editor, *eLife*. The other authors declare that no competing interests exist.

### Funding

| Funder | Grant reference number | Author |
|---|---|---|
| National Institute of Neurological Disorders and Stroke | F31NS092126 | Adriane G Otopalik |
| National Institute of Neurological Disorders and Stroke | R35NS097343 | Eve Marder |
| Grass Foundation | | Adriane G Otopalik |

The funders had no role in study design, data collection and interpretation, or the decision to submit the work for publication.

### Author contributions
Adriane G Otopalik, Conceptualization, Resources, Data curation, Software, Formal analysis, Funding acquisition, Investigation, Visualization, Methodology, Writing—original draft, Writing—review and editing; Jason Pipkin, Software, Formal analysis, Investigation, Visualization, Methodology, Writing—original draft, Writing—review and editing; Eve Marder, Conceptualization, Resources, Supervision, Funding acquisition, Writing—original draft, Writing—review and editing

### Author ORCIDs
Adriane G Otopalik http://orcid.org/0000-0002-3224-6502
Jason Pipkin http://orcid.org/0000-0001-5525-3951
Eve Marder https://orcid.org/0000-0001-9632-5448

### Decision letter and Author response
Decision letter https://doi.org/10.7554/eLife.41728.030
Author response https://doi.org/10.7554/eLife.41728.031

## Additional files

### Supplementary files
• Transparent reporting form
DOI: https://doi.org/10.7554/eLife.41728.025

### Data availability
All computational scripts used for: analysis, visualization, and model simulations are available on the Marder Lab GitHub site (https://github.com/marderlab/Otopalik-Pipkin-Marder-2019), where they are freely available to the public. All source data (electrophysiology recordings and raw images) are publicly available on Dryad (https://dx.doi.org/10.5061/dryad.48pt6jd).

The following dataset was generated:

| Author(s) | Year | Dataset title | Dataset URL | Database and Identifier |
|---|---|---|---|---|
| Adriane G Otopalik, Jason Pipkin, Eve Marder | 2019 | Data from: Neuronal morphologies built for reliable physiology in a rhythmic motor circuit | https://dx.doi.org/10.5061/dryad.48pt6jd | Dryad Digital Repository, 10.5061/dryad.48pt6jd |

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
