## [Decision Letter]

Thank you for submitting your article "Neuronal morphologies built for compact computing in a rhythmic motor circuit" for consideration by *eLife*. Your article has been reviewed by two peer reviewers, and the evaluation has been overseen by a Reviewing Editor and Ronald Calabrese as the Senior Editor. The following individuals involved in review of your submission have agreed to reveal their identity: Gilles Laurent (Reviewer #1).

Summary:

This nice paper is the continuation of a series of studies from the Marder lab investigating the functional consequences of STG neuron morphology in the passive regime. The authors combine an exploration of parameter space by compartmental modeling with experimental tests, contrasting dendritic integration in neurites with large taper (as in STG neurons) with integration in dendrites with low taper (as is typical of cortical neurons, often used as "typical" or representative examples). The basic claim is that the geometry of the dendrites in this system makes the neurons act virtually as a single compartment even though they have a very elaborate dendritic structure. The authors claim that this is a result of substantial tapering of the dendrites and conduct compartmental simulations to support their conclusion. The paper is well written and the experimental sample size impressive. It seems to do a good job in considering and testing different cases with simulation and matching experimental data to simulations which is the gold standard in this type of studies. Nevertheless, we ask the authors to address several concerns, as listed below, to strengthen the conclusions of this work.

Essential revisions:

1) Reading the manuscript gives the feeling a negative result (synaptic inputs seem to be location independent) is turned into a feature. However, there may be methodological reasons why location dependence of some features is found to be weak (see below) and even if it is true, it does not justify consideration as a special mechanism. There are many neurons even in mammalian CNS where apparently there is not much dendritic integration (e.g. cerebellar granular cells).

2) Figure 1 – the authors perform many simulations to test the effect of change in diameter on voltage attenuation along the dendrites. They make the point that geometry can affect attenuation along the dendrites even when all other parameters are kept the same.

a) While the results are presented as a surprise, the understanding that dendritic geometry affects voltage attenuation is at the heart of cable theory and is present already in Rall's 59 paper.

b) In addition, the specific case of tapering is analytically covered in the papers of Schierwagen (admittedly very difficult to read): Schierwagen, A.K., A non-uniform equivalent cable model of membrane voltage changes in a passive dendritic tree, J. Theor. Biol. (1989) 141, 159-179, which is not even cited.

c) The simulated case assumes sealed end boundary condition at both sides of the cable, while in fact the relevant case is that there is a large "load" (e.g. killed end boundary condition) on the side with large diameter, because the rest of the dendritic tree is connected to that part. This might have a significant effect on the conclusions.

d) The authors have both the experimental data of the geometry of the neurons (which they studied in detail in the first paper) and the physiological properties (allowing them to extract the biophysical properties of the membrane). They could simulate the specific neuron they record from, rather than choose to simulate simplified models, which might fail to capture the intricate properties of the real geometry.

3) The authors use sharp electrode recordings and in fact two of them at the soma, which indeed makes the current clamp better. However, it is accepted that sharp electrode recordings may fundamentally change the estimate of membrane parameters and may create a significant conductance leak in the recording location. On top of that, most of the relevant integration happens near the location of synaptic inputs and far away from the recording site. So, it is difficult to escape the alternative explanation in which all the input looks similar at the recording point because they all very far from it, especially when the inputs are very slow (0.5 s rise time, voltage attenuation is far smaller for steady state inputs as compared to transients, and here the inputs are so slow that they are virtually steady state).

4) The authors find a fit for the data with effective space constant of order of magnitude of 1mm with total dendritic length of > 10mm. This means that there are certainly points that are quite far from each other, and still they claim that the neuron acts as a single compartment.

5) The measurements of E_rev_ in their hands shows almost no sensitivity to dendritic location of the activation. This together with simulation that shows that under certain condition (short space constant) E_rev_ estimation should be sensitive to location is taken as an indication that the space constant is long. However, for this the authors are only using very simplified models, which we suspect are very different in terms of boundary conditions than their experimental setup (see above).

6) The individual responses presented in Figure 5—Figure supplements 1, 2 and especially 3, seem to have different shape indices (i.e. rise time and decay time, consistent with classical cable theory of inputs arriving from different locations along the dendrites) and inconsistent with a single compartment scenario.

---

## [Author Response]

Essential revisions:1) Reading the manuscript gives the feeling a negative result (synaptic inputs seem to be location independent) is turned into a feature. However, there may be methodological reasons why location dependence of some features is found to be weak (see below) and even if it is true, it does not justify consideration as a special mechanism. There are many neurons even in mammalian CNS where apparently there is not much dendritic integration (e.g. cerebellar granular cells).

We have adjusted the text of the Results and Discussion sections to clearly state the predictions generated by the proof-of-concept computational simulations and how these predictions relate to the experimental results. We hope that it is now clear that the experimental results align well with the predictions generated by the passive cable simulations. This is not a negative result, but an excellent outcome. The alignment of the experimental and simulation results tells us that the compact electrotonic structures and voltage integration properties observed in these four neuron types can potentially arise from passive voltage propagation in the absence of active properties, in part because they present such wide neurite diameters. This is stated clearly in the Discussion:

“…we find that multiple STG neuron types present electrotonically compact structures and within-neurite voltage summation that is relatively linear and directionally insensitive. […] Our computational simulations suggest that this biophysical architecture may be achieved passively and as a simple consequence of neurite geometry in these neurons, which present secondary branches with wide diameters that taper from 10-20 µm at their proximal, primary-neurite junctions, to sub-micron diameters at their terminating tips.”

Although the reviewer suggests that there are many neuron types thought not to perform compartmentalized, dendritic computations, the direct experimental assessment of electrotonus and such compartmentalization has actually only been executed in a handful of neuron types. We describe this in the Introduction:

To date, measuring voltage attenuation across the many neurite paths presented in complex neuronal structures using electrophysiological techniques has proven difficult. Thus, electrotonus has been experimentally assessed in only a handful of neuron types (for example: Spruston and Johnston, 1992; Spruston et al., 1994; Rapp et al., 1994; Carnevale et al., 1997; Stuart and Spruston, 1998; Chitwood et al., 1999; Jaffe and Carnevale, 1999; Otopalik et al., 2017b; Medan et al., 2018), and this greatly restricts our understanding of the breadth of biophysical organizations utilized in different neuron types and circuit contexts.

While we understand that STG neurons may not be the first neuron types to be described as electrotonically compact, we do think it important to present this study, along with the founding papers (Otopalik et al. 2017a, b), as a case wherein the complex morphologies are in fact compact, and this is perhaps what allows for robust circuit output in the face of a great deal of morphological variability in the same neuron types across animals.

2) Figure 1 – the authors perform many simulations to test the effect of change in diameter on voltage attenuation along the dendrites. They make the point that geometry can affect attenuation along the dendrites even when all other parameters are kept the same.a) While the results are presented as a surprise, the understanding that dendritic geometry affects voltage attenuation is at the heart of cable theory and is present already in Rall's 59 paper.

Our description of the simulation results as ‘surprising’ was not meant to be interpreted as ‘novel.’ This said, we have altered the text throughout the manuscript to clarify this. Our computational simulations in NEURON rely heavily on the seminal theoretical findings of Rall, Schierwagen, and many others. Our intention in presenting these simulations was to apply these seminal theories, generate predictions, and provide a useful pedagogical narrative throughout the paper (Goldstein, 2018). Yet, it is also important to distinguish the analytical approaches of these seminal studies from our numerical simulations: whereas these earlier studies generated biophysical rules and cable equations, in our simulations we were able to scan a broad parameter space of passive and morphological properties and test the boundaries of these aforementioned rules regarding electrotonus, directional sensitivity, and passive voltage integration. By doing this, we found that the wide neurite diameters exhibited by STG neurons renders them relatively resilient to the conventional electrotonic decrement expected from such long neurite paths. The fact that this phenomenon is predicted by passive cable models in NEURON, suggests that our experimental observations can indeed be explained by passive propagation and neurite geometry. By revising the last paragraph of the Introduction, we believe we will have prepared the reader to enter the Results section with this outlook.

b) In addition, the specific case of tapering is analytically covered in the papers of Schierwagen (admittedly very difficult to read): Schierwagen, A.K., A non-uniform equivalent cable model of membrane voltage changes in a passive dendritic tree, J. Theor. Biol. (1989) 141, 159-179, which is not even cited.

We appreciate that this study was brought to our attention and regret that it was not on our radar at the time of writing this manuscript. One hurdle faced in pursuing this work is synthesizing and making sense of the wealth of previous studies examining geometry, electrotonus, and neuronal function. The diverse numerical, analytical, and experimental methods used in previous studies have led to a somewhat incongruous literature. This said, we have now read Schierwagen (1989) closely. In this study, Schierwagen derived the equations for equivalent cable models for multiple branched trees or subtrees. These were then used to determine the distribution of subthreshold membrane voltage in highly branched neuronal structures. This is now cited in the Introduction (first paragraph) and again in the Results section (subsection “Simulating Electrotonus in Diverse Neurites”, first paragraph). It is important to note that, in this study, Schierwagen considered taper as a descriptor of an equivalent cylinder for multiply branched trees or subtrees; the taper we refer to in our manuscript is simply referring to the geometry of individual neurite paths, not multiply branched trees. Additional studies we failed to reference in the original manuscript, that consider the role of taper and flare of individual branches (Goldstein and Rall, 1974; Holmes and Rall, 1992), have now been properly cited in the subsection 2 Linking Electrotonus and Neurite Geometry in STG Neurons”.

c) The simulated case assumes sealed end boundary condition at both sides of the cable, while in fact the relevant case is that there is a large "load" (e.g. killed end boundary condition) on the side with large diameter, because the rest of the dendritic tree is connected to that part. This might have a significant effect on the conclusions.

This issue that is also brought up in essential revision #5 (below). In this work, we have simulated neurite paths that we view as a representation of the entire path from terminal neurite tip to a recording electrode at the soma. These two ends are, in fact, sealed. However, it is true that there may be some shunting at branch points along this path, particularly at the junction between the secondary and primary neurites. To test how such putative shunting may influence our measurements of electrotonus, we ran a set of cable simulations with a large load on the proximal side of the cable (with the larger diameter, d_0_) in a representative cross-section of our cable model library (R_m_ = 10000 Ω × cm^2^,d_1_ = 0.5 µm (constant); d_0_ = 0.5, 1, 5, 10, 20 µm; R_a_ = 50, 100, 150, 200, 300 Ω × cm). We added the shunt by simply adding a compartment at 300 µm from this proximal end of the parent cable (passive properties consistent with the parent cable), effectively mimicking the rest of the neurite tree. We varied the size of the shunt by simply varying the length of this added compartment (as choosing an exact leak conductance magnitude or cable length was somewhat arbitrary; the different shunt sizes had lengths 100, 300, 500, 1000 µm; all had an axial diameter of 5 µm). We measured the effective λ in these cables, as in Figure 1 and Figure 1—figure supplement 1. We found that increasing the shunt did indeed alter the effective λ measured, but that the relative relationships across the varying cable geometries shown in Figure 1 and Figure 1—figure supplement 1 remained true (that is, increasing the proximal diameter to 10 microns resulted in effective λ greater than 1 mm for a range of axial resistance values). The results of this simulation are now presented in Figure 1—figure supplement 2, and discussed briefly in the Results text (subsection “Simulating Electrotonus in Diverse Neurites”, last paragraph).

d) The authors have both the experimental data of the geometry of the neurons (which they studied in detail in the first paper) and the physiological properties (allowing them to extract the biophysical properties of the membrane). They could simulate the specific neuron they record from, rather than choose to simulate simplified models, which might fail to capture the intricate properties of the real geometry.

In the first submission of this manuscript, path lengths were the only geometrical measurements that we had for the evaluated neuronal structures. In previous work, we had shown that the apparent reversal potentials measured across many sites in GM neurons were independent of distance from soma, neurite diameter, and branch order (Otopalik et al., 2017). Thus, for this paper we simply used ImageJ to measure distance from soma for each photo-activated site, and set aside these other geometric measurements. Thus, in the first submission we lacked other geometrical data for these neurons.

To address this and other concerns expressed by the reviewers in this revised manuscript, we have completed volumetric reconstructions and continuous measurement of neurite diameter from primary neurite to terminating tips of 23 neurites from neurons for which we had high-quality confocal stacks with sufficient resolution (now described in the Materials and methods section “Volumetric Neurite Reconstructions”). In these analyses, we found that, while some neurites tapered gradually from the proximal primary neurite junction to terminal tip, others decreased in diameter in surprisingly abrupt steps. We have summarized these data in Figure 2D-E and presented all diameter measurements in Figure 2—figure supplements 1 and 2. These results are discussed in the subsection “Linking Electrotonus and Neurite Geometry in STG Neurons”.

Having made these finer measurements, we then completed a new set of simulations in a revised set of cable models that test the influence of taper, versus step-reductions, in cable models with varying diameters. We found that, for cables with the wide diameters presented by STG neurites, there exists a regime of passive properties that are resilient to step-reductions in diameter. These results are now shown in Figure 2—figure supplement 3 and discussed in the aforementioned subsection.

3) The authors use sharp electrode recordings and in fact two of them at the soma, which indeed makes the current clamp better. However, it is accepted that sharp electrode recordings may fundamentally change the estimate of membrane parameters and may create a significant conductance leak in the recording location. On top of that, most of the relevant integration happens near the location of synaptic inputs and far away from the recording site. So, it is difficult to escape the alternative explanation in which all the input looks similar at the recording point because they all very far from it, especially when the inputs are very slow (0.5 s rise time, voltage attenuation is far smaller for steady state inputs as compared to transients, and here the inputs are so slow that they are virtually steady state).

We appreciate that sharp recordings are not the standard in many biological preparations, whereas they are the standard in the crustacean stomatogastric ganglion preparation. This is, in part, because STG neurons present relatively large somata (125.8 ± 47.5 µm in diameter; Otopalik et al., 2017a) that are well-suited for sharp recordings with one or two electrodes. In the present study, experiments were only conducted in cells with inputs resistances > 5 MOhms following the insertion of both electrodes, which is consistent with the input resistance cut-off specified in other STG studies, even those utilizing only a single electrode. The kinetics of the events described here are consistent with the glutamate-evoked potentials conducted in cultured, transplanted STG neurons (e.g. Cleland and Selverston, 1995). These transplanted neurons are thought to act essentially as a single compartment, as they grow few neurites in vitro. Furthermore, we and others who work in the STG have now provided increasing evidence that these neurons are relatively compact and that voltage decrement must be minimal for these inhibitory glutamate-mediated voltage events:

1) In this and previous work (Otopalik et al., 2017b), we have demonstrated that we can inject current at the soma and alter the membrane potential at distal activation sites sufficiently enough to flip the sign of the evoked glutamate response at somatic membrane potentials that are within close range of the predicted reversal potential based on the chloride-dependence of the evoked current (Otopalik et al., 2017b; Figure 3 and associated supplements in this manuscript).

2) Summation of events activated in sequence across branches appears relatively linear (Figure 5) and independent of direction of activation. This is consistent with our interpretation of compact electrotonus; dendrites that present a greater degree of electrotonic decrement are thought to give rise to a centripetal bias in voltage propagation and current flow toward the open end of the cable (Barlow and Levick, 1965; Euler et al., 2002; London and Häusser, 2005; Branco et al., 2010). In our study, it appears that the wide diameters of STG neurites render them impervious to such directional biases.

3) Although technically difficult, in previous studies experimentalists have completed dual recordings with one electrode at the soma and a second electrode several hundred microns away on the primary or secondary neurites (Golowasch and Marder, 1992; Miller, 1980). In both accounts, they found that while action potentials are subject to attenuation across this distance, graded inhibitory potentials with kinetics consistent with the events evoked in the present study do not undergo much attenuation. It was not clear from these studies, however, whether the many neurite paths in these complex neuronal structures were uniformly electrotonically compact, or if this was a specific feature of this experimentally accessible path from primary neurite to soma.

4) The authors find a fit for the data with effective space constant of order of magnitude of 1mm with total dendritic length of > 10mm. This means that there are certainly points that are quite far from each other, and still they claim that the neuron acts as a single compartment.

In the Abstract and Introduction (second paragraph) we write that STG neurons often present total cable lengths > 10 mm. The total cable length is the summed length of the entire neuronal structure, not the length of single neurite branches. Because individual secondary branches are typically no longer than 1mm, events arising at disparate locations are unlikely to be greater than 2 mm apart. A lambda of ~1 mm would certainly allow for integration of events evoked at sites separated by this distance. To ensure that this confusion does not arise among readers, we have clarified the meaning of total cable length in the Introduction. We now write: “expansive and complex neurite trees that sum to > 10 mm of total cable length…”.

5) The measurements of E_rev_ in their hands shows almost no sensitivity to dendritic location of the activation. This together with simulation that shows that under certain condition (short space constant) E_rev_ estimation should be sensitive to location is taken as an indication that the space constant is long. However, for this the authors are only using very simplified models, which we suspect are very different in terms of boundary conditions than their experimental setup (see above).

This is addressed in our response to comment 2, part c.

6) The individual responses presented in Figure 5—figure supplements 1, 2 and especially 3, seem to have different shape indices (i.e. rise time and decay time, consistent with classical cable theory of inputs arriving from different locations along the dendrites) and inconsistent with a single compartment scenario.

The kinetics of the events described here are consistent with the glutamate-evoked potentials conducted in cultured, transplanted STG neurons (Cleland and Selverston, 1995). Yet, it is true that there is heterogeneity in shape (as described by the reviewer) and amplitude across sites. Given the uniformity of apparent reversal potentials and the long lambdas predicted with the cable simulations in neurites with geometries similar to those observed in these neurons, we are inclined to speculate that these variable shape indices arise from local variations in receptor densities across photo-activated sites (this is discussed in Otopalik et al., 2017b).